# So-Fake: Benchmarking Social Media Image Forgery Detection

## Abstract

Recent advances in AI-powered generative models have enabled the creation of increasingly realistic synthetic images, posing significant risks to information integrity and public trust on social media platforms. While robust detection frameworks and diverse, large-scale datasets are essential to mitigate these risks, existing academic efforts remain limited in scope: current datasets lack the diversity, scale, and realism required for social media contexts, and evaluation protocols rarely account for explanation or out-of-domain generalization. To bridge this gap, we introduce **So-Fake**, a comprehensive social media-oriented dataset for forgery detection consisting of two key components. First, we present **So-Fake-Set**, a large-scale dataset with over **2 million** photorealistic images from diverse generative sources, synthesized using a wide range of generative models. Second, to rigorously evaluate cross-domain robustness, we establish **So-Fake-OOD**, a novel and large-scale (**100K**) out-of-domain benchmark sourced from real social media platforms and featuring synthetic imagery from commercial models explicitly excluded from the training distribution, creating a realistic testbed that mirrors actual deployment scenarios. Leveraging these complementary datasets, we present **So-Fake-R1**, a baseline framework that applies reinforcement learning to encourage interpretable visual rationales. Experiments show that So-Fake surfaces substantial challenges for existing methods. By integrating a large-scale dataset, a realistic out-of-domain benchmark, and a multi-dimensional evaluation protocol, So-Fake establishes a new foundation for social media forgery detection research.

## 1 Introduction

The rapid evolution of generative AI (Shuai et al., 2024; Hu et al., 2025) has made it increasingly difficult to verify the authenticity of social media images, as it enables malicious actors to create deceptive content that misleads public opinion or spreads false information. This has motivated the creation of large-scale datasets to study and improve forgery detection. In recent years, several deepfake datasets (Yan et al., 2024; Bhattacharyya et al., 2024; Zhu et al., 2023) have been proposed to train more robust forgery detection models. However, they generally exhibit three significant limitations that make them inadequate for addressing the complex challenges of social media image forgery detection: **1) Narrow Categorical Scope.** Existing datasets (Ricker et al., 2024; Peng et al., 2024; Yang et al., 2023) focus narrowly on specific categories such as faces, animals, or humans, failing to represent complex real-world social media contexts. **2) Outdated Generation Quality.** Most datasets (Zhu et al., 2023; Corvi et al., 2023; Huang et al., 2024b) rely on outdated generation techniques, which result in less convincing forgeries that are easier for both humans and models to detect. **3) Limited Cross-Domain Evaluation.** Existing datasets lack established protocols for measuring cross-domain generalization and rarely include a dedicated out-of-distribution benchmark. While recent works (Ricker et al., 2024; Huang et al., 2025b; Dell'Anna et al., 2025) have attempted to introduce forgery detection datasets for social media images, they face significant constraints in acquiring authentic platform content. Existing benchmarks approximate social media imagery indirectly, for instance by re-uploading generated images or substituting generic open datasets. These proxies fail to capture the fidelity, compression, and topical diversity of authentic social media content, highlighting the need for datasets that more faithfully reflect real-world conditions.

Beyond dataset limitations, existing evaluation protocols also remain inadequate. Social media forgeries range from fully synthetic to regionally tampered images (Huang et al., 2025b), which

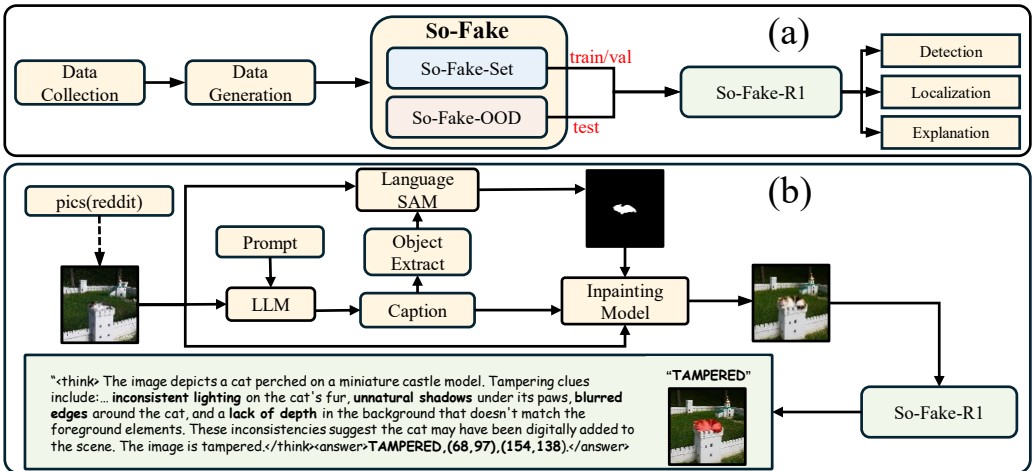

Figure 1: **(a) Overview.** So-Fake comprises So-Fake-Set (train/val) and So-Fake-OOD (test), which together enable evaluation of detection, localization, and explanation with So-Fake-R1. **(b) Illustrative Example.** A real image from the subreddit *pics* is captioned by an LLM, combined with Language SAM and an inpainting model to produce tampered samples. So-Fake-R1 then analyzes the manipulated image and outputs the class label, localized region, and an interpretable rationale.

calls for benchmarks that evaluate not only detection but also manipulation localization and explanation to foster user trust. Despite progress, most methods remain limited: many target only face deepfakes as binary classification (Yan et al., 2024; 2023; Kroiß & Reschke, 2025), or extend to mask prediction (Dong et al., 2023; Guo et al., 2023; Zhang et al., 2024), yet provide little insight into their decisions. This black-box nature further limits applicability in social media, where diverse manipulations demand transparent explanations. To address this challenge, recent advances in large language models (Li et al., 2025; Kang et al., 2025; Xu et al., 2024; Ji et al., 2025) have inspired explainability methods that generate human-readable rationales, but these depend on costly human annotations and cannot jointly address tampered and fully synthetic cases. Consequently, existing methods and protocols remain inadequate for social media forgeries, highlighting the need for joint evaluation of detection, localization, and explanation to ensure trustworthiness.

To address these limitations, we introduce **So-Fake**, a comprehensive benchmark for social media forgery detection with explicit protocols for evaluating detection, localization, and explanation. As illustrated in Figure 1, it consists of two complementary components. **So-Fake-Set** is the main training and validation corpus, comprising over **2M** images across **12** diverse categories (see Figure 2 (a) and Figure 3 (a)) and extending beyond traditional face-centric datasets to include humans, animals, and events. **So-Fake-OOD** is a **100K**-image out-of-distribution benchmark that pairs authentic social media content from Reddit[1] with synthetic imagery generated by leading commercial models listed in the Leaderboard[2] . The **30** generation and manipulation methods in So-Fake-Set are entirely disjoint from the **10** commercial models in So-Fake-OOD (see Figure 2 (b)), reflecting the closed-source nature of many real-world tools and enabling realistic evaluation of generalization to unseen generators. In both datasets, images are labeled as real, tampered, or full synthetic, reflecting the major forms of fake content encountered on authentic social media environments.

Leveraging these complementary datasets, we further provide **So-Fake-R1**, a baseline framework that illustrates the practical use of So-Fake for social media forgery detection. So-Fake-R1 leverages reinforcement learning (RL) to produce interpretable predictions, enabling comprehensive evaluation across detection, localization, and explanation. The main contributions of this paper are as follows:

- We introduce **So-Fake**, a large-scale social media-oriented dataset comprising **So-Fake-Set** for training/validation and **So-Fake-OOD** for out-of-distribution evaluation.
- We propose **So-Fake-R1**, an RL-based framework that unifies detection, localization, and explanation of social media forgeries, thereby demonstrating the utility of So-Fake.
- Extensive experiments demonstrate So-Fake's effectiveness as a comprehensive benchmark, with So-Fake-R1 achieving state-of-the-art results across detection, localization, and explanation tasks while maintaining strong generalization to out-of-distribution domains.

---

[1]https://www.reddit.com
[2]https://artificialanalysis.ai/text-to-image

Table 1: Comparison with recent image forgery datasets. "–" in #Methods indicates the number of generative methods was not specified; "–" in Latest Fake indicates the specific generative method was not specified; Column abbreviations: MultiCls = Multiclasses, Expl. = Explanation.

| Dataset | Social Media | Latest Fake | #Methods | Data Sources | #Images | MultiCls | Mask | Expl. | OOD |
|---|---|---|---|---|---|---|---|---|---|
| ArtiFact ('23) | ✗ | Palette ('22) | 25 | COCO, FFHQ, LSUN | 2M+ | ✗ | ✗ | ✗ | ✗ |
| DMimage ('23) | ✗ | DALL-E ('22) | 10 | COCO, ImageNet | 0.4M+ | ✗ | ✗ | ✗ | ✗ |
| AIGCD ('23) | ✗ | Wukong ('22) | 16 | LSUN, COCO, FFHQ | 0.7M+ | ✗ | ✗ | ✗ | ✗ |
| SynthScars ('25) | ✗ | FLUX ('24) | - | RichHF-18K, Chameleon, FFAA | 13K | ✗ | ✓ | ✓ | ✗ |
| FakeClue ('25) | ✗ | FLUX ('24) | - | GenImage, FF++, Chameleon | 0.1M+ | ✗ | ✗ | ✓ | ✗ |
| GenImage ('23) | ✗ | Wukong ('22) | 8 | ImageNet | 2M+ | ✗ | ✗ | ✗ | ✗ |
| WildFake ('25) | ✗ | DALL-E 3 ('23) | 27 | ImageNet, COCO, FFHQ, LSUN, +3 more | 3M+ | ✗ | ✗ | ✗ | ✗ |
| Community Forensics ('25) | ✗ | FLUX ('24) | 4803 | LAION, ImageNet, COCO, FFHQ, +7 more | 2M+ | ✗ | ✗ | ✗ | ✓ |
| SID-Set ('24) | ✓ | FLUX ('24) | 2 | COCO, Flickr30k, MagicBrush | 0.3M | ✓ | ✓ | ✓ | ✗ |
| Deepfake-Eval-2024 ('25) | ✓ | - | - | X, Tiktok, Instagram | 1975 | ✗ | ✗ | ✗ | ✗ |
| TrueFake ('25) | ✓ | FLUX ('24) | 8 | FFHQ, FORLAB, Facebook, X, Telegram | 0.6M+ | ✗ | ✗ | ✗ | ✗ |
| **So-Fake** | ✓ | Nano Banana ('25) | 40 | COCO, Flickr30k, WIDER, OpenForensics Reddit, OpenImages, FFHQ, CelebA | 2M+ | ✓ | ✓ | ✓ | ✓ |

## 2 RELATED WORK

### 2.1 IMAGE FORGERY DETECTION DATASETS

Early datasets such as DFFD (Dang et al., 2020), ForgeryNet (He et al., 2021), and FaceForensics++ (Rössler et al., 2019) established the foundation for deepfake detection research, albeit with a narrow emphasis on GAN-generated facial forgeries (Karras et al., 2020; 2021a). With the rise of diffusion models, research has expanded beyond facial manipulations to encompass broader AI-generated content (AIGC) detection. This trend is reflected in the emergence of recent benchmarks such as GenImage (Zhu et al., 2023) and DMimage (Corvi et al., 2023). As detection tasks increasingly target content circulating in real-world environments, attention has shifted towards constructing specialized datasets for social media forgery detection (Ricker et al., 2024; Huang et al., 2025b; Dell'Anna et al., 2025). Despite these advances, current datasets exhibit notable shortcomings, including reliance on outdated generative techniques and insufficient diversity in real-world scenarios. More recently, WildFake (Hong et al., 2025) collected millions of community-shared synthetic images from platforms, while Community Forensics (Park & Owens, 2025) systematically sampled from open-source and commercial generators, achieving unprecedented model coverage. However, both datasets emphasize open repositories rather than real social media distributions, and neither provides multi-class labels, tampering masks, or explanations. In contrast, **So-Fake** is the first dataset explicitly targeting **social media forgeries**, with two distinctive advantages: (1) social-media-oriented data collection rather than relying on open repositories or community uploads; and (2) enriched annotations and benchmarks, including multi-class labels, tampered region masks, explanatory rationales, and a dedicated OOD split based on real social media data for rigorous cross-domain evaluation. A detailed comparison with existing image forgery datasets is provided in Table 1.

### 2.2 IMAGE FORGERY DETECTION, LOCALIZATION, AND EXPLANATION

Recent developments in forgery detection have focused primarily on using deep neural networks to distinguish authentic content from manipulated content. While these methods (Kroiß & Reschke, 2025; Chen et al., 2022; Pei et al., 2024; Wang et al., 2025b) achieve strong performance by capturing subtle visual artifacts, they often lack robustness when facing novel manipulation types or content domains. To address these limitations, researchers have increasingly turned to localization approaches that identify specific tampered regions. Image forgery detection and localization (IFDL) (Dong et al., 2023; Guo et al., 2024; Guillaro et al., 2023; Zhang et al., 2023; Liu et al., 2022) provides more granular and interpretable insights than global classification alone, enabling a better understanding of manipulation techniques. However, current localization datasets focus almost exclusively on facial forgeries (Wang et al., 2025a; Liang et al., 2024), neglecting the diverse manipulation types characteristic of social media images. In parallel with addressing data limitations, interpretability has emerged as a critical frontier, with recent approaches attempting to generate human-understandable justifications alongside detection. Motivated by recent vision-language models, several works, such as ForgeryGPT (Li et al., 2024b), SIDA (Huang et al., 2025b), FakeShield (Xu et al., 2024), FakeScope (Li et al., 2025), and LEGION (Kang et al., 2025) can generate explanations, they typically require extensive manual annotations and produce superficial descriptions that fail to reveal genuine model reasoning. In contrast to these approaches, So-Fake-R1 applies reinforcement learning to

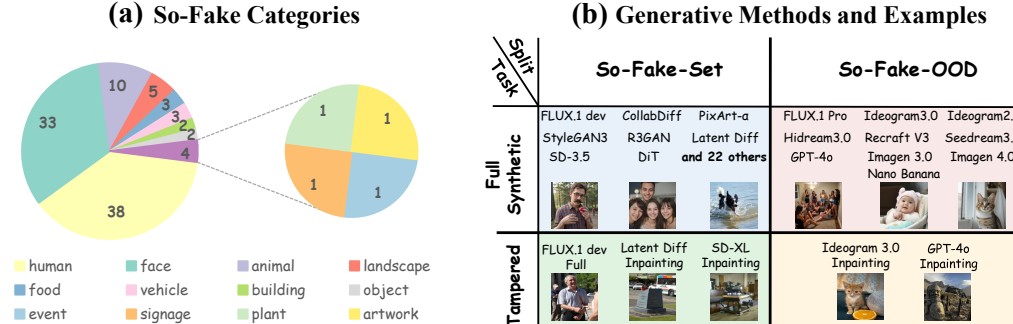

Figure 2: (a) Category distribution of So-Fake across 12 semantic classes. (b) Generative methods and examples for each split–task combination.

enhance the consistency and informativeness of model outputs, enabling detection, localization, and explanation within a unified framework, while reducing reliance on manual annotations.

# 3 DATASET

## 3.1 OVERVIEW

Social media platforms host vast volumes of user-generated images that differ substantially from standard academic datasets. Unlike curated benchmarks, these images cover highly diverse and informal content categories (Dell'Anna et al., 2025), often contain compression artifacts or mixed editing traces (Cozzolino et al., 2024; Huang et al., 2025b), and are increasingly interspersed with synthetic media generated by commercial models such as GPT-4o (Hurst et al., 2024), Hidream3.0 (HiDream-ai, 2025), and Imagen4 (Google, 2025a). These factors make forgery detection in social media particularly challenging, as models must generalize across heterogeneous, noisy, and manipulation-rich distributions, highlighting the need for realistic benchmarks. However, collecting and releasing large-scale authentic social media imagery is severely constrained by platform policies and privacy concerns, making it infeasible to construct a fully open benchmark directly from these sources.

To address these challenges, we propose **So-Fake**, a benchmark explicitly designed for social-media-oriented forgery detection. So-Fake consists of two complementary components: (i) **So-Fake-Set**, a controlled in-domain benchmark, and (ii) **So-Fake-OOD**, for cross-domain robustness testing. Both components are built under a unified 12-class taxonomy spanning humans, objects, events, and natural scenes (Figure 2(a)), ensuring broad semantic coverage representative of social media content. Specifically, **So-Fake-Set** combines diverse public datasets with systematically generated forgeries, providing an open and scalable alternative to unreleasable social media data. The complete list of generative models used in So-Fake-Set is provided in Appendix B.1. **So-Fake-OOD** integrates authentic Reddit images with forgeries synthesized by entirely disjoint commercial generative models, thereby introducing realistic distributional shifts for robust generalization testing (Figure 2(b)).

## 3.2 DATA COLLECTION

**So-Fake-Set.** We select real images from COCO (Lin et al., 2014), Flickr30k (Plummer et al., 2017), OpenImages (Benenson & Ferrari, 2022) and WIDER (Xiong et al., 2015), as these datasets contain complex scenes with humans, animals, diverse environments, and daily activities typical of social media content. We also incorporate CelebA (Liu et al., 2015), OpenForensics (Le et al., 2021), and FFHQ (Karras et al., 2021b) to ensure comprehensive coverage of facial content, which constitutes a significant portion of social media imagery. In total, So-Fake-Set comprises approximately 650K real images, 650K fully synthetic samples, and 650K tampered samples, as shown in Figure 3(a) (left).

**So-Fake-OOD.** For the OOD benchmark, we collect images from Reddit via its official API. Reddit provides diverse, informal user-generated content across our 12 predefined categories, but with styles and quality levels that differ markedly from the open datasets used in So-Fake-Set, creating a realistic domain shift (illustrated in Appendix C.1). Importantly, Reddit's content policy permits non-commercial academic use, ensuring legal compliance (Appendix E). From this collection, we retain around 33K images as real samples, some of which are further used to generate full synthetic and tampered counterparts, resulting in 33K per class. This design enables evaluation under both real image shifts and generative method shifts, as demonstrated in Figure 3(a) (right).

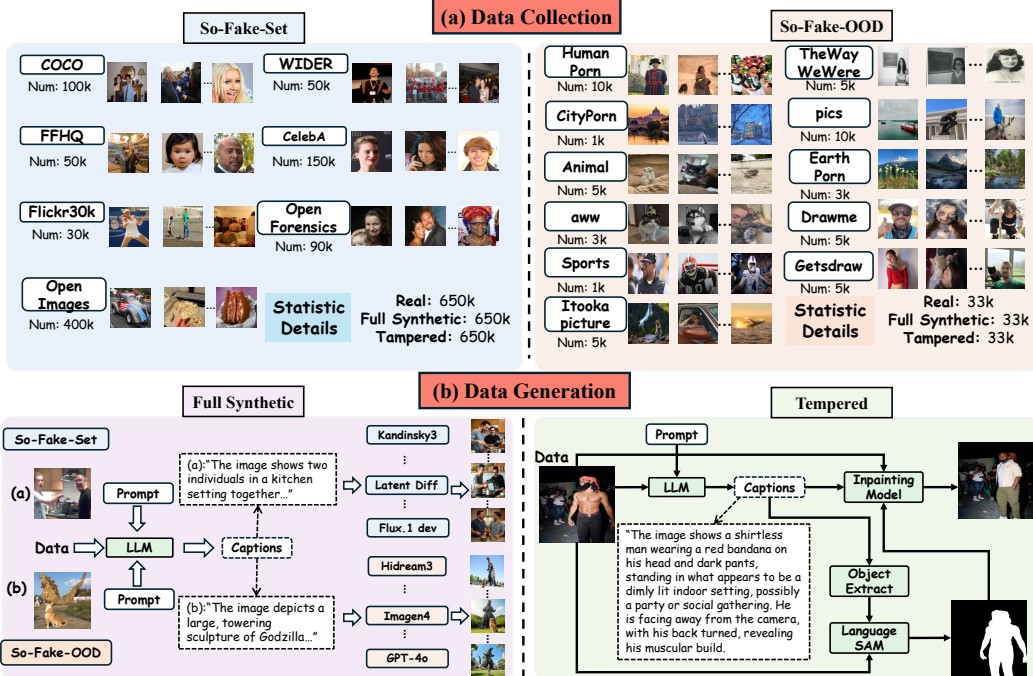

Figure 3: (a) Data collection sources of So-Fake-Set and So-Fake-OOD with representative examples and statistics. (b) Data generation pipelines for full synthetic and tampered images.

## 3.3 DATA GENERATION

We generate two types of synthetic images: **full synthetic** and **tampered**. Figure 3 (b) illustrates their corresponding generation pipelines, which we detail below.

**Full Synthetic Images.** To produce full synthetic images, we adopt two main categories of generation techniques: **GAN-based** and **diffusion-based**. For GAN-based methods, we follow the official implementation guidelines provided by the authors. For diffusion-based approaches, we employ a text-to-image generation pipeline, divided into two stages: **(1) Caption Generation.** We evaluated several captioning models for generating text-to-image captions and selected Qwen2.5-VL-7B (Bai et al., 2025) based on experimental results (Appendix B.2). **(2) Image Synthesis.** The generated captions are then fed into different sets of generative models. For So-Fake-Set, we employ 30 architectures spanning both GAN-based and diffusion-based paradigms, selected to maximize architectural diversity and category coverage (see Appendix B.1). For So-Fake-OOD, we instead adopt state-of-the-art commercial generators, including Hidream 3.0 (HiDream-ai, 2025), Nano Banana (Google, 2025b), and Imagen 4 (Google, 2025a), as summarized in Figure 2(b). **(3) Quality Assessment.** Finally, all generated outputs from both GAN-based and diffusion-based methods are subjected to the same quality control process. We combined automated filtering with human evaluation: five expert reviewers assessed generated samples on realism, consistency, and appropriateness using a five-point scale, removing low-quality images after secondary review. This process guarantees that both So-Fake-Set and So-Fake-OOD maintain consistently high quality. Further details are provided in Appendix B.3, with an illustrative example in Figure 7.

**Tampered Images.** To simulate partial forgeries common in social media imagery, we employ image inpainting techniques that replace specific regions while preserving the rest of the original image. For So-Fake-Set, we use three state-of-the-art inpainting models, including FLUX.1-Fill-dev (Labs, 2024), Latent Diffusion (Rombach et al., 2022), and Stable Diffusion-XL (Podell et al., 2024), chosen for their high visual quality and diversity of generative styles. We first use Qwen2.5-VL-7B (Bai et al., 2025) to generate captions for the source images. An Object Extract module then identifies candidate entities (e.g., "man") from these captions using a lightweight NLP parser. The extracted object labels are passed to LangSAM (lang-sam team, 2024) to generate precise masks for the corresponding regions. Each inpainting model subsequently receives three inputs—the original image, the extracted mask, and the caption—ensuring semantic consistency between the replaced regions and their surrounding context. For So-Fake-OOD, we adopt the same tampering pipeline as So-Fake-Set but replace the inpainting models with GPT-4o (Hurst et al., 2024), and

Ideogram 3.0 (Ideogram, 2025), ensuring evaluation on unseen manipulation techniques. This design simultaneously introduces distribution shift in real images and manipulation methods, yielding a more realistic OOD benchmark.

# 4 METHOD

## 4.1 OVERVIEW

We introduce **So-Fake-R1** as a unified vision–language policy optimization baseline for our benchmark. It formulates forgery detection as a multi-objective reinforcement learning problem, where Group Relative Policy Optimization (GRPO) is used to align three complementary goals: detection, localization, and explanation. The two-stage pipeline first establishes a stable reasoning format through cold-start supervised tuning, and then refines the model with GRPO

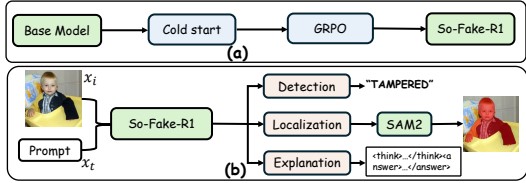

Figure 4: (a) Training pipeline with SFT and GRPO. (b) Inference pipeline producing detection, localization, and explanation outputs.

to jointly enhance task synergy, as illustrated in Figure 4 (a). This design offers two advantages: (i) balanced optimization, where gains in one task do not compromise others, and (ii) structured interpretability guided by rule-based rewards, reducing reliance on additional human supervision.

## 4.2 TRAINING

**Stage 1: Cold start.** We fine-tune the base model on a curated set of 9,000 images from So-Fake-Set (balanced across the three classes, with annotations generated by GPT-4o and subsequently refined through strict expert review; see Appendix D.2) to align it with our reasoning format and structured output requirements. This initialization is essential for teaching the model consistent formatting and multi-granular reasoning, without which reinforcement learning would fail to converge reliably. The dataset is intentionally kept small to prevent overfitting while still providing a strong prior for subsequent optimization, reinforced by structured reasoning cues.

**Stage 2: GRPO training.** Building on this initialization, we apply GRPO to refine the model jointly across detection, localization, and explanation. Unlike cold-start stage, which only provides static supervision, GRPO incorporates rule-based rewards that capture complementary objectives and encourage balanced improvements across tasks. This stage is particularly critical for harmonizing reasoning quality with localization precision, while avoiding reliance on manual annotations.

As shown in Section 5.5, the two stages provide complementary gains and yield the best overall performance. Detailed specifications of the cold-start dataset construction, reward weighting strategy, and training hyperparameter choices are provided in the Appendix D.2.

## 4.3 REWARD FUNCTIONS

The reward function is designed to align with the three core outputs of So-Fake-R1: explanation, detection, and localization. Accordingly, we group the components into three categories, each providing complementary signals. **(1) Explanation format rewards.** To encourage interpretable reasoning, we adopt a *format reward* that enforces structured outputs. Specifically, explanations must appear inside `<think>...</think>` and final answers inside `<answer>...</answer>` tags. This ensures parseable reasoning traces and stabilizes optimization. **(2) Detection reward.** We assign a reward based on the correctness of the predicted label (REAL, TAMPERED, FULL_SYNTHETIC), extracted from the `<answer>` tags. Correct predictions for REAL or FULL SYNTHETIC images receive a base reward. For TAMPERED images, which typically involve subtle and localized manipulations and are thus harder to detect, a higher reward is assigned. This weighting prevents optimization from being biased toward the easier classes and ensures balanced training across all three categories. **(3) Localization rewards.** For images predicted as TAMPERED, we provide additional rewards to enforce precise localization: (a) a *format reward* ensures that bounding boxes follow a strict coordinate specification; (b) an *IoU reward* grants a positive score when predicted boxes achieve Intersection-over-Union (IoU) $> 0.5$ with the ground truth; (c) an *L1 reward* further

refines accuracy by rewarding predictions within 10 pixels of the ground truth coordinates. For REAL and FULL_SYNTHETIC images, default values are assigned to maintain balanced gradient signals across all classes. The total reward combines these signals:

$$R_{\text{total}} = \lambda_{\text{fmt}}R_{\text{fmt}} + \lambda_{\text{cls}}R_{\text{cls}} + \lambda_{\text{seg\_fmt}}R_{\text{seg\_fmt}} + \lambda_{\text{IoU}}R_{\text{IoU}} + \lambda_{\text{L1}}R_{\text{L1}}, \tag{1}$$

where $\lambda$ controls the relative weight of each component. This decomposition ensures that So-Fake-R1 is jointly optimized for structured reasoning, reliable detection, and fine-grained localization. The exact reward weights, scoring criteria, and implementation details are reported in the Appendix D.1.

### 4.4 INFERENCE PROCEDURE

At inference time, So-Fake-R1 takes as input an image $x_i$ together with a task prompt $x_t$ and produces three complementary outputs, as illustrated in Figure 4(b). **Explanation:** a hierarchical reasoning trace enclosed in `<think>...</think>`, which analyzes the image at different perspectives before reaching a conclusion. **Detection:** a class label (REAL, TAMPERED, or FULL_SYNTHETIC) reported within `<answer>...</answer>` tags. **Localization:** for tampered cases, So-Fake-R1 outputs bounding-box coordinates in the format `<|box_start|>(x1, y1), (x2, y2)<|box_end|>|`. These coarse boxes are then passed to SAM2 (Ravi et al., 2024), which refines the bounding boxes into dense segmentation masks.

## 5 EXPERIMENT

### 5.1 EXPERIMENTAL SETTINGS

**Methods.** For a fair and comprehensive comparison of So-Fake-R1 across detection, localization, and explanation tasks, we evaluate against three representative groups of baselines: **(1) Detection-only methods.** CnnSpot (Frank & Holz, 2021), UnivFD (Ojha et al., 2023), FreAware (Tan et al., 2024b), and NPR (Tan et al., 2024a). **(2) Image Forgery Detection and Localization (IFDL) methods.** HIFI-Net (Guo et al., 2024), TruFor (Guillaro et al., 2023), PSCC-Net (Liu et al., 2022), SIDA (Huang et al., 2025b), and FakeShield (Xu et al., 2024). **(3) Explanation-oriented methods.** LLaVA-1.5-13B (Liu et al., 2023), LISA (Lai et al., 2024), InternVL3-8B (Zhu et al., 2025), Qwen2.5-VL-7B (Bai et al., 2025), and DeepSeek-VL-7B (DeepSeek-AI et al., 2025). Unless otherwise noted, all baselines are fine-tuned on the So-Fake-Training Set. Exceptions include: FakeShield, which requires paired image-text inputs and is evaluated using its publicly released checkpoints; and HIFI-Net, which is evaluated using its pre-trained weights due to the unavailability of complete training code.

**Metrics.** We evaluate models across the three tasks defined by So-Fake. For **detection**, we report image-level accuracy (Acc) and F1. For **localization**, we adopt Intersection over Union (IoU) and mask-level F1, which capture the ability to pinpoint subtle local edits that are prevalent in social-media manipulations. For **explanation**, we employ two complementary metrics: (1) Cosine Semantic Similarity (CSS), which captures semantic alignment between embeddings of predicted and ground-truth explanations, and (2) ROUGE-L, which quantifies textual overlap through longest common subsequence matching. Ground-truth explanations were first generated with Claude Opus 4.1 (light logo, 2025), then carefully revised and validated by human experts. In total, over 3,000 high-quality explanations were curated, providing reliable supervision for quantitative benchmarking.

### 5.2 COMPARISON RESULTS ON SO-FAKE-SET

As shown in Table 2, our method achieves superior performance across all metrics, surpassing the second-best method by **1.3%** in detection accuracy, **1.1%** in localization IoU, and significantly higher CSS scores for explanation quality. These results demonstrate the effectiveness of So-Fake-R1.

### 5.3 COMPARISON RESULTS ON SO-FAKE-OOD

For fairness, we evaluate both the *zero-shot* and *fine-tune* settings, where models are fine-tuned on the training split of So-Fake-Set and evaluated on So-Fake-OOD. As shown in Table 3, So-Fake-R1 achieves the highest performance across all metrics, demonstrating superior cross-domain generalization compared to other methods.

Table 2: Performance comparison on So-Fake-Set. Methods marked with "*" denote results obtained using publicly released weights without fine-tuning.

| Method | Year | Type | Detection | | Localization | | Explanation | |
|---|---|---|---|---|---|---|---|---|
| | | | Acc | F1 | IOU | F1 | ROUGE-L | CSS |
| CnnSpot | 2021 | Detection | 89.6 | 87.7 | - | - | - | - |
| UnivFD | 2023 | Detection | 84.0 | 63.8 | - | - | - | - |
| FreAware | 2024 | Detection | 85.6 | 73.1 | - | - | - | - |
| NPR | 2024 | Detection | 81.8 | 61.5 | - | - | - | - |
| HIFI-Net* | 2022 | IFDL | 39.0 | 25.2 | 12.1 | 18.3 | - | - |
| TruFor | 2023 | IFDL | 87.3 | 85.9 | 47.5 | 57.6 | - | - |
| PSCC-Net | 2022 | IFDL | 84.2 | 81.1 | 46.3 | 54.8 | - | - |
| FakeShield* | 2024 | IFDL | 67.0 | 64.1 | 33.7 | 46.1 | 0.2412 | 0.5143 |
| SIDA | 2025 | IFDL | 91.9 | 91.5 | 44.1 | 58.9 | 0.4313 | 0.7987 |
| LLaVA-1.5-13B | 2024 | LLM | 83.5 | 82.9 | 29.8 | 38.1 | 0.4213 | 0.7877 |
| LISA | 2024 | LLM | 87.4 | 85.9 | 40.5 | 47.6 | 0.4246 | 0.7861 |
| DeepSeek-VL-7B | 2025 | LLM | 83.7 | 81.1 | 27.8 | 35.4 | 0.4376 | 0.8196 |
| Qwen2.5-VL-7B | 2024 | LLM | 91.2 | 90.8 | 42.7 | 50.1 | 0.4515 | 0.8411 |
| InternVL3-8B | 2025 | LLM | 87.6 | 87.3 | 41.1 | 48.5 | 0.4553 | 0.8341 |
| Ours | 2025 | LLM | **93.2** | **92.9** | **48.6** | **63.9** | **0.4718** | **0.8769** |

Table 3: Performance comparison on So-Fake-OOD with both *zero-shot* and *fine-tune* settings.

| Method | Detection | | | | Localization | | | | Explanation | | | |
|---|---|---|---|---|---|---|---|---|---|---|---|---|
| | zero shot | | fine tune | | zero shot | | fine tune | | zero shot | | fine tune | |
| | Acc | F1 | Acc | F1 | IOU | F1 | IOU | F1 | ROUGE-L | CSS | ROUGE-L | CSS |
| CnnSpot | 32.8 | 29.7 | 65.2 | 63.8 | - | - | - | - | - | - | - | - |
| UnivFD | 45.3 | 43.7 | 63.3 | 40.2 | - | - | - | - | - | - | - | - |
| FreAware | 52.3 | 48.3 | 56.5 | 54.6 | - | - | - | - | - | - | - | - |
| NPR | 55.6 | 47.1 | 57.6 | 50.9 | - | - | - | - | - | - | - | - |
| HIFI-Net* | 54.3 | 47.3 | - | - | 15.2 | 22.4 | - | - | - | - | - | - |
| TruFor | 44.7 | 12.6 | 55.9 | 53.1 | 7.8 | 11.2 | 32.3 | 41.1 | - | - | - | - |
| PSCC-Net | 35.4 | 9.9 | 48.9 | 46.1 | 20.5 | 30.7 | 41.1 | 48.7 | - | - | - | - |
| FakeShield* | 42.1 | 35.7 | - | - | 24.9 | 30.2 | - | - | 0.2561 | 0.5214 | - | - |
| SIDA | 50.1 | **49.8** | 73.1 | 72.2 | **25.4** | **38.9** | 40.1 | 49.3 | 0.1724 | 0.4026 | 0.4135 | 0.7899 |
| LLaVA-1.5-13B | 34.1 | 33.2 | 70.9 | 70.5 | 9.7 | 13.8 | 26.7 | 35.1 | 0.1026 | 0.3321 | 0.4212 | 0.7689 |
| LISA | 37.6 | 37.1 | 70.1 | 70.0 | 18.3 | 21.4 | 38.2 | 47.5 | 0.1663 | 0.4106 | 0.4115 | 0.7881 |
| DeepSeek-VL-7B | 35.9 | 34.0 | 71.1 | 70.4 | 10.3 | 14.1 | 25.4 | 34.6 | 0.1054 | 0.3422 | 0.4212 | 0.7776 |
| Qwen2.5-VL-7B | 38.4 | 35.7 | 73.3 | 72.5 | 17.5 | 20.3 | 42.2 | 49.9 | **0.2692** | 0.5342 | 0.4371 | 0.8124 |
| InternVL3-8B | 39.1 | 33.6 | 71.2 | 70.1 | 10.2 | 13.8 | 40.4 | 47.1 | 0.2653 | **0.5473** | 0.4463 | 0.8231 |
| Ours | - | - | **76.4** | **75.3** | - | - | **47.8** | **59.1** | - | - | **0.4695** | **0.8421** |

## 5.4 EXTERNAL EXPERIMENTS

**Robustness Evaluation.** We evaluate So-Fake-R1's robustness against common social media perturbations, including JPEG compression (quality 70/80), resizing (scale 0.5/0.75), and Gaussian noise (variance 5/10). As shown in Table 4, our model maintains strong performance across all degradation scenarios, demonstrating its practical applicability for real-world deployment.

**Evaluation on External Social Media Benchmark.** To further assess generalization capabilities, we evaluate So-Fake-R1 on SID-Set (Huang et al., 2025b). As shown in Table 5, So-Fake-R1 demonstrates strong cross-dataset generalization after fine-tuning.

**Cross-Domain Generalization Analysis.** We analyze the generalization capabilities across different generators to understand the challenges posed by So-Fake. As shown in Figure 5, detectors generalize reasonably within So-Fake-Set when training and testing generators share architectural traits, but performance drops sharply across distinct paradigms. In particular, cross-family transfer between GAN and diffusion models is noticeably weaker, and this gap becomes far more pronounced under So-Fake-OOD. This suggests that current detection methods may be overfitting to generator-specific artifacts rather than learning fundamental patterns. Additional analysis is provided in Appendix C.

## 5.5 ABLATION STUDY

**Training Strategy.** We evaluate the effectiveness of our two-stage training pipeline. As shown in Table 6, both the cold-start SFT stage and GRPO refinement contribute to final performance. The cold-start stage is crucial for establishing basic detection capabilities, while GRPO significantly improves all metrics. Without cold-start training, the model struggles to properly identify tampered content categories, demonstrating that GRPO alone is insufficient for our challenging detection tasks.

Table 4: Performance of So-Fake-R1 under different perturbations.

|  | Detection | | Localization | | Explanation | |
| --- | --- | --- | --- | --- | --- | --- |
|  | ACC | F1 | F1 | IOU | ROUGE-L | CSS |
| JPEG 70 | 91.5 | 91.2 | 60.3 | 45.1 | 0.4523 | 0.8612 |
| JPEG 80 | 92.0 | 91.8 | 61.7 | 46.3 | 0.4611 | 0.8658 |
| Resize 0.5 | 89.7 | 89.1 | 58.4 | 42.9 | 0.4352 | 0.8483 |
| Resize 0.75 | 90.9 | 90.3 | 59.8 | 44.2 | 0.4477 | 0.8540 |
| Gaussian 10 | 88.3 | 87.5 | 54.7 | 40.5 | 0.4124 | 0.8306 |
| Gaussian 5 | 89.8 | 89.0 | 56.1 | 41.7 | 0.4239 | 0.8407 |
| Original | 93.2 | 92.9 | 63.9 | 48.6 | 0.4718 | 0.8769 |

Table 5: Comparison of So-Fake-R1 and other deepfake detection methods on SID-Set.

| Methods | Real | | Fake | | Overall | |
| --- | --- | --- | --- | --- | --- | --- |
|  | Acc | F1 | Acc | F1 | Acc | F1 |
| CnnSpott | 89.0 | 90.8 | 79.4 | 76.1 | 84.2 | 83.5 |
| Gram-Net | 89.2 | 91.7 | 93.9 | 92.8 | 91.6 | 92.3 |
| Fusing | 89.2 | 92.7 | 57.6 | 60.3 | 73.4 | 76.5 |
| UnivFD | 68.3 | 68.5 | 89.5 | 94.0 | 78.9 | 81.3 |
| AntifakePromp | 88.9 | 89.1 | 94.2 | 89.2 | 91.6 | 89.2 |
| SIDA-7B | 89.1 | 91.0 | 95.0 | 94.8 | 92.1 | 92.9 |
| Ours | **91.1** | **92.9** | **95.6** | **95.1** | **93.4** | **94.0** |

Table 6: Training strategy.

| Cold Start | GRPO | Detection Acc. |
| --- | --- | --- |
|  | ✓ | 63.7 |
| ✓ |  | 89.3 |
| ✓ | ✓ | **93.2** |

Table 7: Reward function.

| $R_{cls}$ | $R_{fmt}$ | $R_{seg\_fmt}$ | $R_{IOU}$ | $R_{L1}$ | Acc | IOU |
| --- | --- | --- | --- | --- | --- | --- |
| ✓ |  |  |  |  | 93.1 | - |
| ✓ | ✓ |  |  |  | 93.1 | - |
| ✓ | ✓ | ✓ |  |  | 92.9 | - |
| ✓ | ✓ | ✓ | ✓ |  | 93.2 | 46.7 |
| ✓ | ✓ | ✓ | ✓ | ✓ | 93.2 | 48.6 |

Table 8: Policy model.

| Policy Model | Detection | |
| --- | --- | --- |
|  | Acc | F1 |
| InternVL3-8B | 91.3 | 90.8 |
| DeepSeek-VL-7B | 88.6 | 88.1 |
| Qwen2.5-VL-7B | **93.2** | **92.9** |

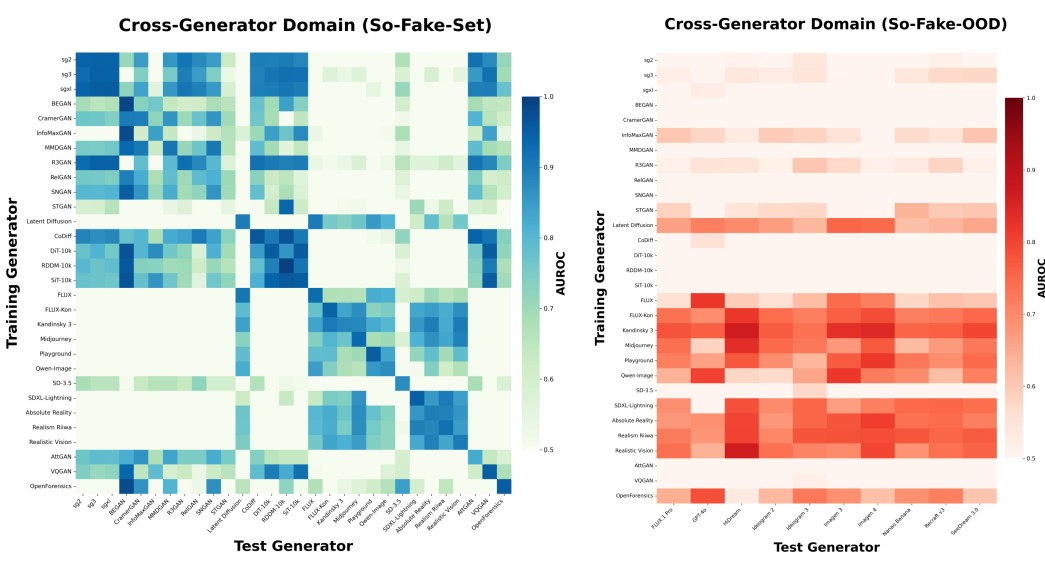

Figure 5: Cross-generator domain generalization matrix using CNNSpot. Rows indicate training generators and columns indicate test generators, with So-Fake-Set (left) and So-Fake-OOD (right).

**Selection of Reward Functions.** We analyze the impact of different reward function combinations on model performance. As shown in Table 7, using all five reward functions yields the best results.

**Policy Model Selection.** We evaluate several policy models, and Table 8 shows that Qwen2.5-VL-7B achieves the best performance, supporting our final choice.

## 6 CONCLUSION AND LIMITATION

**Conclusion.** We present **So-Fake**, a benchmark for social-media forgery detection that includes **So-Fake-Set** for training/validation and **So-Fake-OOD** for cross-domain evaluation. We further propose **So-Fake-R1**, an RL-based framework that unifies detection, localization, and explanation, offering a strong baseline in both in-domain and OOD settings. **Limitations.** While So-Fake advances the scale and diversity of forgery benchmarks, it still cannot fully capture the breadth of real-world social media content or the rapidly evolving landscape of generative models. So-Fake-R1, though effective, remains computationally demanding and may produce inaccurate localization in challenging cases, as shown in Appendix D.4. Finally, our benchmark focuses on still images, leaving video and multimodal forgeries as important directions for future work.

**Ethics Statement.** This work adheres to ethical research practices detailed throughout our paper. Data collection from Reddit follows their public content policy for non-commercial academic use (Appendix E), with multi-stage human review processes to filter inappropriate content (Appendix B.3). Our research aims to advance detection capabilities against malicious synthetic media, with datasets and models released exclusively for academic research under controlled access. We acknowledge the dual-use nature of generative AI research and commit to responsible disclosure practices to minimize potential misuse while maximizing societal benefits through improved detection capabilities.

**Reproducibility Statement.** We provide comprehensive implementation details to ensure full reproducibility. Complete dataset construction pipelines are detailed in Section 3 and Appendix B, including all 40 generative methods (Table 9), human evaluation protocols (Section B.3), and quality control procedures (Figure 6). Training configurations, hyperparameters, and reward function specifications for So-Fake-R1 are provided in Appendix D.1. Baseline implementations follow official guidelines where available, with detailed configurations in Section D.2. We commit to releasing the So-Fake dataset, So-Fake-R1 source code, trained model weights, and evaluation protocols to facilitate future research and enable direct comparison with our results.

**LLM Usage Statement.** We declare that large language models (LLMs) were used exclusively for language editing and stylistic improvements in this manuscript. They did not contribute to the conceptual, methodological, or experimental aspects of the work.

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

# A    APPENDIX

**Contents of the Appendix:**

Section B provides full details on dataset construction, including:

- The generative methods used in this paper (Section B.1),

- The selection of caption models (Section B.2),

- Human expert assessment, including source image filtering, model choice, and generative quality evaluation (Section B.3),

- Construction of the cold start dataset (Section B.4),

- The process of generating the ground-truth explanations (Section B.5).

Section C presents benchmark analyses, including:

- Dataset statistics and distributional comparisons (Section C.1),

- Cross-domain generalization analysis (Section C.2),

- Category-level and forgery-type asymmetry studies (Section C.3),

- Representative dataset samples (Section C.4),

- Duplicate detection analysis (Section C.5).

Section D outlines experimental settings and method details, including:

- Reinforcement learning configuration (Section D.1),

- Implementation settings (Section D.2),

- Qualitative comparison on tampered cases (Section D.3),

- Additional qualitative examples (Section D.4).

Section E discusses the broader social impact of our work.

# B    DATA CONSTRUCTION

In this section, we present the construction pipeline of the So-Fake dataset. We first describe the generative methods employed to synthesize both fully synthetic and tampered images in So-Fake-Set and So-Fake-OOD (Section B.1). We then detail the selection of caption models used to provide textual descriptions images (Section B.2). Next, we outline the human expert assessment procedure, including source filtering, model validation, and quality control (Section B.3). We further introduce the cold start dataset designed to initialize reasoning capabilities with structured annotations (Section B.4). Finally, we describe the generation of explanation ground-truths, which serve as reliable supervision for evaluating model outputs (Section B.5).

## B.1    GENERATIVE METHODS

We describe the generative models and prompting strategies used to synthesize both fully synthetic and tampered images in So-Fake-Set and So-Fake-OOD, as shown in Table 9. In total, we employ 40 generative methods to ensure the dataset is comprehensive and diverse. To maintain architectural balance, we include a range of both GAN-based and diffusion-based models. While some methods are relatively outdated, a limited number are retained to reflect historical trends and enhance robustness. For more advanced models such as FLUX (Labs, 2024), SD-3.5 (Esser et al., 2024), SD-XL (Podell et al., 2024), and Latent Diffusion (Rombach et al., 2022), we generate a larger volume of samples due to their accessibility and high visual fidelity.

Table 9: Details of generative methods used in constructing So-Fake-Set and So-Fake-OOD. Column abbreviations: Set = So-Fake-Set, OOD = So-Fake-OOD, F = fully synthetic images, T = tampered images. Real data source abbreviations: F30k = Flickr30k, OI = OpenImages, OF = OpenForensics.

| ID-Number | Method | Model Type | Data Used | Generation Target | Year | Venue | Real Data Source | Data Scale | Code Link |
|---|---|---|---|---|---|---|---|---|---|
| 1 | Absolute_Reality (Lykon, 2023) | Diffusion | Set | F | 2023 | None | F30k & WIDER | 50000 | Hyper-link |
| 2 | AttGAN (He et al., 2017) | GAN | Set | F | 2017 | PAMI | FFHQ & CelebA | 2000 | Hyper-link |
| 3 | BEGAN (Berthelot et al., 2017) | GAN | Set | F | 2017 | NeurIPS | FFHQ & CelebA | 2000 | Hyper-link |
| 4 | Collaborative Diffusion (Huang et al., 2023) | Diffusion | Set | F | 2023 | CVPR | FFHQ & CelebA | 30000 | Hyper-link |
| 5 | CramerGAN (Bellemare et al., 2017) | GAN | Set | F | 2017 | NeurIPS | FFHQ & CelebA | 2000 | Hyper-link |
| 6 | DiT-XL (Peebles & Xie, 2023) | Diffusion | Set | F | 2023 | ICCV | FFHQ & CelebA | 10000 | Hyper-link |
| 7 | FLUX.1-dev (Labs, 2024) | Diffusion | Set | F & T | 2024 | None | FFHQ & CelebA & OI & COCO & OF | 350000+ | Hyper-link |
| 8 | InfoMaxGAN (Lee et al., 2021) | GAN | Set | F | 2021 | WACV | FFHQ & CelebA | 2000 | Hyper-link |
| 9 | Kandinsky3 (Arkhipkin et al., 2023) | Diffusion | Set | F | 2023 | None | F30k & WIDER | 80000 | Hyper-link |
| 10 | Latent diffusion (Rombach et al., 2022) | Diffusion | Set | F & T | 2022 | CVPR | F30k & WIDER &COCO & OI | 250000+ | Hyper-link |
| 11 | MMDGAN (Li et al., 2017) | GAN | Set | F | 2017 | NeurIPS | FFHQ & CelebA | 2000 | Hyper-link |
| 12 | playground (Li et al., 2024a) | Diffusion | Set | F | 2024 | None | F30k & WIDER | 50000 | Hyper-link |
| 13 | R3GAN (Huang et al., 2024a) | GAN | Set | F | 2024 | NeurIPS | FFHQ & CelebA | 30000 | Hyper-link |
| 14 | RDDM (Liu et al., 2024) | Diffusion | Set | F | 2024 | CVPR | FFHQ & CelebA | 10000 | Hyper-link |
| 15 | rRealism_riiwa (riiwa, 2024) | Diffusion | Set | F | 2024 | None | F30k & WIDER | 50000 | Hyper-link |
| 16 | RelGAN (Nie et al., 2019) | GAN | Set | F | 2019 | ICLR | FFHQ & CelebA | 2000 | Hyper-link |
| 17 | SD-3.5 (Esser et al., 2024) | Diffusion | Set | F | 2024 | ICML | FFHQ & CelebA & OI & COCO & F30k | 100000+ | Hyper-link |
| 18 | SD-XL (Podell et al., 2024) | Diffusion | Set | F & T | 2024 | ICLR | F30k & WIDER & COCO &OF | 300000+ | Hyper-link |
| 19 | StyleGAN-2 (Karras et al., 2020) | GAN | Set | F | 2020 | CVPR | FFHQ & CelebA | 30000 | Hyper-link |
| 20 | StyleGAN-3 (Karras et al., 2021a) | GAN | Set | F | 2021 | NeurIPS | FFHQ & CelebA | 30000 | Hyper-link |
| 21 | StyleGAN-XL (Podell et al., 2024) | GAN | Set | F | 2022 | SIGGRAPH | FFHQ & CelebA | 30000 | Hyper-link |
| 22 | SiT-XL (Ma et al., 2024) | Diffusion | Set | F | 2024 | ECCV | FFHQ & CelebA | 10000 | Hyper-link |
| 23 | SNGAN (Miyato et al., 2018) | GAN | Set | F | 2018 | ICLR | FFHQ & CelebA | 2000 | Hyper-link |
| 24 | STGAN (Liu et al., 2019) | GAN | Set | F | 2019 | CVPR | FFHQ & CelebA | 2000 | Hyper-link |
| 25 | VQGAN (Yu et al., 2022) | GAN | Set | F | 2022 | ICLR | FFHQ & CelebA | 2000 | Hyper-link |
| 26 | OpenForensics (Le et al., 2021) | GAN | Set | F | 2021 | CVPR | Google Open Images | 10000 | Hyper-link |
| 27 | Realistic Vision (Realistic_Vision, 2023) | Diffusion | Set | F | 2023 | None | F30k & WIDER & COCO &OF | 50000 | Hyper-link |
| 28 | FLUX.1-Kontext-dev (Labs, 2024) | Diffusion | Set | F | 2025 | None | OI | 8000+ | Hyper-link |
| 29 | Qwen-image (Wu et al., 2025) | Diffusion | Set | F | 2025 | Arxiv | OI | 3000+ | Hyper-link |
| 30 | Midjourney (Midourney, 2023) | Diffusion | Set | F | 2023 | None | OI | 8000+ | Hyper-link |
| 31 | Recraft-v3 (recraftAI, 2024) | Diffusion | OOD | F | 2024 | None | Reddit | 5000+ | Hyper-link |
| 32 | GPT-4o(OpenAI, 2023) | Diffusion | OOD | F & T | 2025 | Arxiv | Reddit | 8000+ | Hyper-link |
| 33 | Image3 (Baldridge et al., 2024) | Diffusion | OOD | F | 2025 | Arxiv | Reddit | 3000+ | Hyper-link |
| 34 | Image4 (Baldridge et al., 2024) | Diffusion | OOD | F | 2025 | Arxiv | Reddit | 3000+ | Hyper-link |
| 35 | Nano Banana (Baldridge et al., 2024) | Diffusion | OOD | F | 2025 | None | Reddit | 2000+ | Hyper-link |
| 36 | Seedream3.0 (Gao et al., 2025) | Diffusion | OOD | F | 2025 | Arxiv | Reddit | 5000+ | Hyper-link |
| 37 | Ideogram3.0 (Ideogram, 2025) | Diffusion | OOD | F & T | 2025 | None | Reddit | 5000+ | Hyper-link |
| 38 | Ideogram2.0 (Ideogram, 2024) | Diffusion | OOD | F | 2024 | None | Reddit | 5000+ | Hyper-link |
| 39 | FLUX1.1_pro (Labs, 2025) | Diffusion | OOD | F | 2025 | None | Reddit | 3000+ | Hyper-link |
| 40 | Hi-Dream (HiDream-ai, 2025) | Diffusion | OOD | F | 2025 | None | Reddit | 3000+ | Hyper-link |

Table 10: Comparison of image captioning models for prompt generation using CLIP score.

| Method | Mean±Std |
|---|---|
| Qwen2.5-VL-7B | **0.3361±0.034** |
| InternVL2-7B | 0.3258±0.034 |
| BLIP-2 | 0.3047±0.036 |
| InstructBLIP | 0.2996±0.034 |
| LLaVA-7B | 0.2974±0.037 |

### B.2 CAPTION MODEL SELECTION

To ensure high-quality prompts for diffusion-based text-to-image synthesis, we first evaluate multiple captioning models on their ability to produce detailed descriptions of source images. Specifically, we randomly sample 1,000 images from So-Fake-Set and use five popular image captioning LLMs to generate corresponding prompts. We then compare prompt quality using CLIP similarity score, as shown in Table 10. The results show that Qwen2.5-VL-7B (Bai et al., 2025) achieves the highest CLIP score, and we therefore adopt it as our unified prompt generator for So-Fake.

### B.3 HUMAN EXPERT ASSESSMENT

We conducted a rigorous human evaluation to ensure the quality and appropriateness of our dataset. Five experts participated in a multi-stage process that included: (1) filtering source images, (2) validating model selection choices, and (3) assessing the quality of generated synthetic content. The overall process is illustrated in Figure 6.

**Source Image Filtering.** To construct a realistic and diverse OOD benchmark, we carefully selected subreddits based on their popularity, topical diversity, and coverage of broad content categories representative of typical social media platforms. Specifically, we selected 11 subreddits, including (1) `pics`, (2) `HumanPorn`, (3) `aww`, (4) `EarthPorn`, (5) `Getsdraw`, (6) `Drawme`, (7) `TheWayWeWere`, (8) `Animal`, (9) `Ittookapicutre`, (10) `Sports`, and (11) `cityporn`. We

Figure 6: Human expert assessment process comprising **source image filtering**, **model selection**, and **image quality assessment**. This multi-stage evaluation ensures the dataset's authenticity, diversity, and high visual quality for realistic deepfake detection scenarios.

used the official Reddit API[3] to collect images and adhered to Reddit's Public Content Policy[4], ensuring that all images were used strictly for non-commercial research purposes.

After collecting the initial dataset, experts manually reviewed the Reddit images and filtered out those deemed inappropriate, harmful, or irrelevant to social media image analysis. This process resulted in a curated set of 100K high-quality source images for building the So-Fake-OOD benchmark.

**Model Selection.** Five human experts conducted a thorough evaluation of generative models to determine which should be included in So-Fake-Set and So-Fake-OOD. Each candidate model was reviewed based on its visual output quality, prompt controllability, content diversity, and relevance to current generative trends. The experts considered both widely adopted and emerging models, ensuring a balanced mix of GAN-based and diffusion-based architectures. This process resulted in the selection of 40 models, ranging from early GANs like StyleGAN and AttGAN to state-of-the-art diffusion models such as SD-XL (Podell et al., 2024), FLUX (Labs, 2024), and DiT (Peebles & Xie, 2023). By involving domain experts in model evaluation, we ensured that the final selection reflects both high technical quality and representative diversity for real-world image forgery scenarios.

**Image Quality Assessment.** Expert reviewers assessed the realism of generated images using a five-point scale (0–5, with 5 indicating the highest quality). Images scoring below 3 underwent secondary review, and confirmed low-quality samples were excluded from the dataset. For So-Fake-Set, we adopted a two-stage filtering process: an initial automated pass using QwenVL-7B (Bai et al., 2025) to remove clearly non-photorealistic images, followed by random human spot-checks. In contrast, all So-Fake-OOD images were subjected to a comprehensive manual review to ensure consistently high visual quality and realism. An illustrative example is provided in Figure 7.

### B.4    COLD-START DATASET CONSTRUCTION

To initialize the reasoning capabilities of So-Fake-R1, we constructed a dedicated cold-start dataset. Our goal in this stage was not to maximize scale, but to establish a compact yet diverse corpus that provides consistent supervision signals and prevents overfitting.

**Sampling strategy.** We evenly sampled images from So-Fake-Set across all 30 generative methods and 12 semantic categories, ensuring broad coverage of both content and generation styles. In total, we curated approximately 9,000 images, deliberately balanced across the three classes (real, fully synthetic, tampered).

**Annotation process.** For each selected image, we obtained chain-of-thought explanations using GPT-4o (Hurst et al., 2024), chosen for its strong visual reasoning and descriptive capabilities. The prompts

---

[3]https://www.reddit.com/prefs/apps

[4]https://www.reddit.com/r/reddit/comments/1co0xnu/sharing_our_public_content_policy_and_a_new/

| Generator | Analysis | Score |
|---|---|---|
| SiT | Blurry facial edges, missing details. | 2,1,2,2,3 |
| Latent Diffusion | Mostly natural, but noticeable inconsistencies (e.g., cow's shape) | 3,3,4,3,3 |
| Nano Banana | Rich details, overall photorealistic. | 5,4,5,5,5 |

Figure 7: Illustrative examples of generation quality and reviewer scores at low, medium, and high quality levels.

followed a structured template (Listing 1), designed to elicit hierarchical rationales highlighting at least five concrete visual cues. For tampered images, we additionally attached bounding box coordinates derived from the corresponding binary manipulation masks, where the box was computed by extracting the extreme pixel coordinates (top-left and bottom-right) of the altered region. The standardized format `"TAMPERED,<|box_start|>"` $(x_1, y_1), (x_2, y_2)$ `"<|box_end|>"` allows models to explicitly associate textual justifications with localized manipulated regions.

**Quality control.** Although the explanations were generated automatically, we conducted lightweight human verification to ensure semantic consistency and adherence to the prescribed output format. Outputs exhibiting ambiguity or structural errors were removed. This procedure provided a high-confidence set of supervision signals without incurring the cost of large-scale manual annotation.

Listing 1: Prompt templates used for constructing the cold-start dataset.

```
PROMPT_TEMPLATES = {
    "REAL": """<image>

You are a forensic image analyst.

This image is known to be **real** (unaltered and directly from a
    camera).

TASK:
In <= 1024 characters, describe at least five concrete visual
    cues from this specific image that indicate it is authentic
    and unedited.
The first sentence should depict the details of the given image.

OUTPUT FORMAT:
Return exactly two blocks:
```

```
1) <think> ...your entire reasoning here... </think>
2) <answer> REAL </answer>

EXAMPLE:
<think>The image shows a woman with voluminous, side-swept wavy
    hair, wearing makeup and gazing slightly to the right.
    Several visual cues suggest the image is authentic and
    unedited: natural, consistent shadows across the face;
    realistic skin texture and tonal variation; hair strands
    blending smoothly into the background without abrupt edges;
    natural facial depth around the nose and cheekbones; and no
    signs of digital artifacts or unnatural
    blending.</think><answer>REAL</answer>

Now analyze the current image and respond using the exact format
    specified above.
""",

    "TAMPERED": """<image>
<mask>

You are a forensic image analyst.
You will receive two images. The first image has been
    **tampered**. The second image is a binary mask (white =
    altered pixels) showing the tampered area of the first image.

TASK:
In <= 1024 characters, first describe the scene, then describe at
    least five concrete manipulation clues visible around the
    highlighted area (e.g., lighting mismatch, edge artifacts).
Do **not** mention the mask file itself.

OUTPUT FORMAT:
Return exactly two blocks:
1) <think> ...your entire reasoning here... </think>
2) <answer> TAMPERED </answer>

EXAMPLE:
<think>The image shows a crowded subway station where people are
    lined up near the platform doors, with a woman prominently
    posed in front holding a drink. Manipulation clues include a
    lighting mismatch on the woman compared to the ambient scene,
    hard edge artifacts around her silhouette, inconsistent
    shadowing beneath her, sharper resolution than surrounding
    elements, and slight scale distortion making her appear
    unnaturally inserted. The image is
    tampered.</think><answer>TAMPERED</answer>

Now analyze the current image and respond using the exact format
    specified above.
""",

    "FULL_SYNTHETIC": """<image>

You are a forensic image analyst.

This image is known to be **fully synthetic** (entirely generated
    by AI).
```

```
TASK:
In <= 1024 characters, describe at least five visual cues that
    reveal the image was generated artificially.
The first sentence should depict the details of the given image.

OUTPUT FORMAT:
Return exactly two blocks:
1) <think> ...your entire reasoning here... </think>
2) <answer> FULL_SYNTHETIC </answer>

EXAMPLE:
<think>The image shows a shirtless, muscular man with tattoos
    walking down a palm-lined street near parked cars. Telltale
    signs of AI generation include overly smooth skin texture,
    inconsistent lighting between subject and background,
    unnatural tattoo symmetry, subtly distorted vehicle details,
    and unrealistically perfect muscle definition. The image is
    fully synthetic.</think><answer>FULL_SYNTHETIC</answer>

Now analyze the current image and respond using the exact format
    specified above.
"""
}
```

## B.5 EXPLANATION GROUND-TRUTH GENERATION

To supervise the explanation component of So-Fake-R1, we constructed textual ground-truth rationales using a two-step process. First, we employed **Claude Opus 4.1** (light logo, 2025), which was selected due to its strong performance in long-form reasoning and stylistic consistency, to generate chain-of-thought explanations. Each prompt included the original image, the associated tampered region mask (for tampered samples only), and the ground-truth class label (REAL, TAMPERED, or FULL_SYNTHETIC). The model then produced reasoning traces in the required output format.

To ensure semantic reliability, a sample of 3,000 generated explanations was systematically reviewed by human experts. The evaluation applied three criteria:

**(1) Accuracy** – the rationale must correctly describe visual evidence in the image;

**(2) Clarity** – the explanation must be concise and unambiguous;

**(3) Consistency** – reasoning style and conclusions must remain coherent across similar cases.

Outputs that were ambiguous or structurally flawed were refined or discarded. The finalized annotations therefore provide a high-confidence benchmark for evaluating multimodal reasoning. Importantly, these ground-truth explanations are used *only for evaluation* and are distinct from the cold start annotations employed during training (Appendix B.4).

## C  BENCHMARK ANALYSES

In this section, we conduct benchmark analyses to demonstrate the properties and utility of the So-Fake dataset. We begin with dataset statistics and distributional comparisons between So-Fake-Set and So-Fake-OOD (Section C.1). We then analyze cross-domain generalization to assess the challenges introduced by the OOD benchmark (Section C.2). Next, we examine category-level and forgery-type asymmetries to reveal systematic differences across semantic domains and manipulation modes (Section C.3). Finally, we provide representative examples that qualitatively illustrate the data types in So-Fake (Section C.4). Together, these analyses empirically substantiate the scale, diversity, and difficulty of So-Fake.

## C.1 DATASET STATISTICS AND DISTRIBUTIONAL COMPARISONS

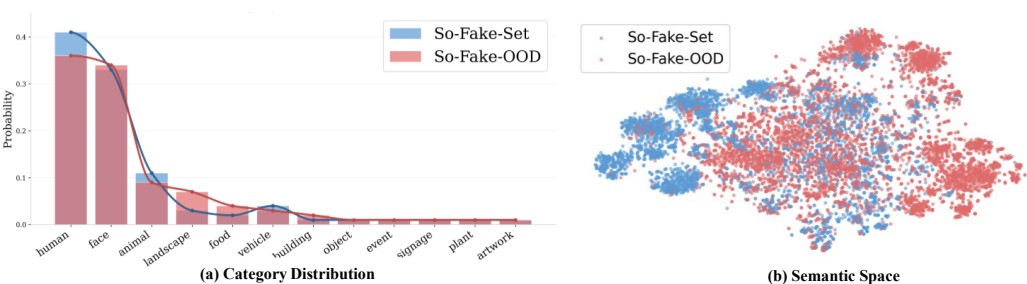

**(a) Category Distribution**      **(b) Semantic Space**

Figure 8: Cross-domain alignment between So-Fake-Set and So-Fake-OOD. (a) Category distribution under the 12-class taxonomy. (b) CLIP embedding visualization.

As outlined in Sec. 3.1, a valid OOD benchmark must maintain comparable semantic coverage while exhibiting measurable distributional shift, particularly in social media contexts. We analyze the relationship between So-Fake-Set and So-Fake-OOD to verify these properties, confirming both semantic alignment and realistic domain gap necessary for robust evaluation.

**Category distribution:** under the unified 12-class taxonomy, the Jensen–Shannon divergence (JSD) is computed on the real subsets to avoid generation-induced artifacts. The divergence is low (JSD $\approx$ 0.08), indicating that So-Fake-OOD provides similar semantic coverage while avoiding category-driven bias in OOD evaluation, as illustrated in Figure 8 (a).

**Semantic space:** CLIP embeddings are extracted from a randomly sampled subset of 10K images from each domain to capture overall semantic coverage. The embedding analysis shows substantial overlap between So-Fake-Set and So-Fake-OOD, while maintaining a measurable domain gap, as illustrate in Figure 8 (b).

Together, these results demonstrate that So-Fake-Set and So-Fake-OOD are semantically aligned yet distributionally distinct, supporting fair and challenging OOD evaluation.

Table 11: Cross-domain generalization performance on So-Fake-OOD using CNNSpot. ID = In-domain; OOD = Out-of-domain.

| Category | Count | ID AUROC | OOD AUROC | Change (%) |
|---|---|---|---|---|
| GAN-based Methods | 14 | 0.594 | 0.396 | -19.8 |
| Open-source Diffusion | 7 | 0.584 | 0.471 | -11.3 |
| Commercial Diffusion | 9 | 0.483 | 0.730 | +24.7 |

## C.2 CROSS-DOMAIN GENERALIZATION ANALYSIS

To systematically analyze cross-domain generalization patterns, we categorize the 30 training methods into three groups based on their technical foundation and deployment context:

**GAN-based models:** BEGAN, CramerGAN, InfoMaxGAN, MMDGAN, RelGAN, SNGAN, STGAN, AttGAN, VQGAN, StyleGAN2, R3GAN, StyleGAN3, StyleGAN-XL, OpenForensics_fake.

**Open-source Diffusion Models:** Latent Diffusion, Collaborative Diffusion, DiT, RDDM, SiT, Stable Diffusion-3.5, Stable Diffusion-XL.

**Commercial / Proprietary Diffusion Models:** FLUX.1-dev, FLUX-1-Kontext-dev, Kandinsky3, Midjourney, Playground, Qwen-image, Absolute_Reality, rRealism_riiwa, Realistic Vision.

Our cross-domain evaluation reveals distinct generalization patterns that reflect the evolving technological landscape of social media forgery. As shown in Table 11 and Figure 5, while GAN-based methods achieve the strongest performance on So-Fake-Set, they exhibit substantial degradation on So-Fake-OOD. Conversely, commercial diffusion models demonstrate the weakest performance

on the training domain but show remarkable cross-domain adaptation, achieving the highest OOD performance. Open-source diffusion models maintain moderate but stable performance.

These analyses capture an important transition in the landscape of social media forgeries. While GAN-based methods remain prevalent in targeted facial manipulations due to their controllability and efficiency, diffusion-based commercial models increasingly dominate the broader ecosystem by producing diverse, high-fidelity content across multiple categories. This technological stratification gives rise to asymmetric detection challenges: detectors trained primarily on academic GAN generators often underperform on emerging commercial diffusion models, whereas detectors exposed to commercial models exhibit more robust generalization across heterogeneous forgery sources.

The generalization gap between commercial and open source diffusion models reflects fundamental differences in their development contexts. Commercial generators face diverse real-world deployment pressures and continuous adversarial challenges in social media environments, leading to distributional properties that enhance detector generalizability. Open source diffusion models, while providing controlled experimental conditions and architectural diversity essential for research, may encode generator-specific artifacts that limit cross-domain transfer.

Importantly, although GAN-based models demonstrate limited OOD generalization, their inclusion remains essential for comprehensive evaluation. These models continue to drive many facial manipulation techniques prevalent on social media platforms, provide crucial contrast to diffusion-based methods that illuminates paradigm shifts in generative technology, and represent persistent threats that detection systems must handle in practice. Taken together, these findings underscore that future detection strategies should move beyond isolated, single-paradigm benchmarks. Effective systems require balanced coverage across both GAN-based and diffusion-based forgeries, supported by adaptive training protocols that account for the diverse threat landscape encountered in real-world deployment scenarios. By capturing the performance divergence between different generative paradigms, So-Fake provides a benchmark that reflects the complex and evolving nature of social media forgeries.

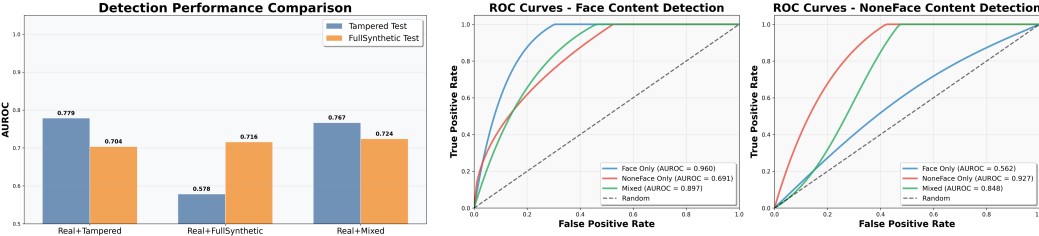

Figure 9: AUROC on tampered vs. full synthetic tests under different settings.

Figure 10: ROC curves for face (left) and none face (right) tests.

### C.3 CATEGORY-LEVEL AND FORGERY-TYPE ASYMMETRY STUDIES

In this section, we move beyond quantitative benchmarks to discuss two key aspects surfaced by So-Fake: (1) differences between tampered and fully synthetic forgeries and (2) the contrast between face-centric and non-face content. We report results using **CNNSpot** as the detector, given its broad adoption as a generator-agnostic baseline.

**Tampered vs. Full Synthetic.** Figure 9 reveals a significant transfer detection gap between tampered and fully synthetic forgeries. Detectors trained on tampered content perform well in-domain (0.779) but performed relatively poorly on synthetic data (0.704), while synthetic-trained models show the reverse pattern (0.578 vs. 0.716). This asymmetry suggests fundamentally different forensic signatures rather than a shared distribution. Mixed training bridges this gap (0.767 vs. 0.724), demonstrating that a unified three-class approach effectively captures both manipulation types while preserving generalization performance. This finding directly validates our design choice for social media forgery detection, where the coexistence of both manipulation types in user feeds necessitates robust cross-category generalization. Our three-class taxonomy (real, tampered, fully synthetic) mirrors the spectrum of fake content prevalent on social platforms, where subtle regional manipulations (e.g., localized inpainting) and entirely AI-generated posts coexist and require unified detection frameworks to ensure comprehensive coverage of real-world scenarios.

**Face vs. Non-Face Forgeries.** Deepfake detection has traditionally centered on facial manipulations, which constitute the majority of existing benchmarks and detection frameworks. To assess whether this focus has created domain-specific biases, we examine detection performance across face and non-face categories on our dataset. Our analysis reveals a clear asymmetry: face-centric forgeries are substantially easier to detect than non-face ones, as shown in Figure 10. Specifically, models achieve an AUROC of 0.960 on face content but only 0.562 on non-face content when trained exclusively on faces, demonstrating a dramatic 39.8% performance drop. However, our integrated approach combining faces with 11 diverse non-face categories yields improved overall robustness—mixed training achieves balanced performance with face detection (AUROC = 0.897) and non-face detection (AUROC = 0.848), representing a 28.6% improvement over face-only training on non-face content. This design choice directly reflects social media reality, where users encounter heterogeneous content spanning portraits, landscapes, objects, and events within the same feeds. By explicitly covering this full spectrum, So-Fake provides a more realistic testbed that addresses the overlooked dimension of generalizable forgery detection across diverse content types.

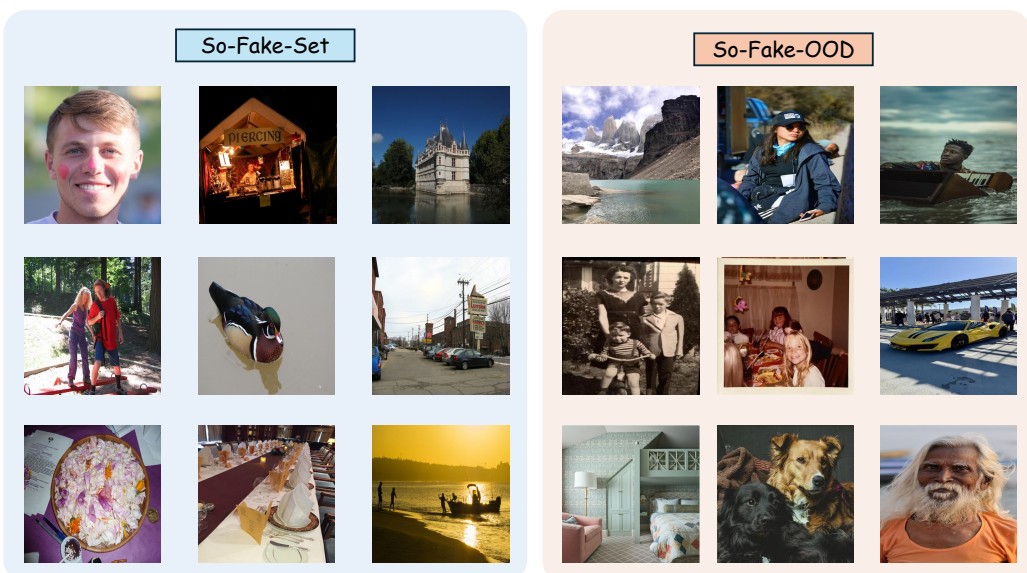

Figure 11: Representative *real* images from So-Fake-Set (left) and So-Fake-OOD (right).

### C.4 REPRESENTATIVE DATASET SAMPLES

In this section, we present representative samples from **So-Fake-Set** and **So-Fake-OOD**, covering the three categories of our benchmark: *real* images from authentic sources, *tampered* images generated by localized inpainting with masks, and *fully synthetic* images created by GANs and diffusion-based models. Examples are shown in Figure 11 (real images), Figure 12 (tampered images), and Figure 13 (fully synthetic images).

### C.5 DUPLICATE DETECTION ANALYSIS

To verify the independence between So-Fake-Set and So-Fake-OOD, we conducted comprehensive similarity analysis using DINOv3 (ViT-L/16) Siméoni et al. (2025), a state-of-the-art self-supervised vision model widely adopted for duplicate detection tasks. We evaluate duplicate similarity under the following configuration:

**Data sampling.** Due to computational constraints, we randomly sample 10% of the real images from each split (seed = 42), resulting in 65,000 sampled real images from So-Fake-Set (650K total) and 3,300 from So-Fake-OOD (33K total).

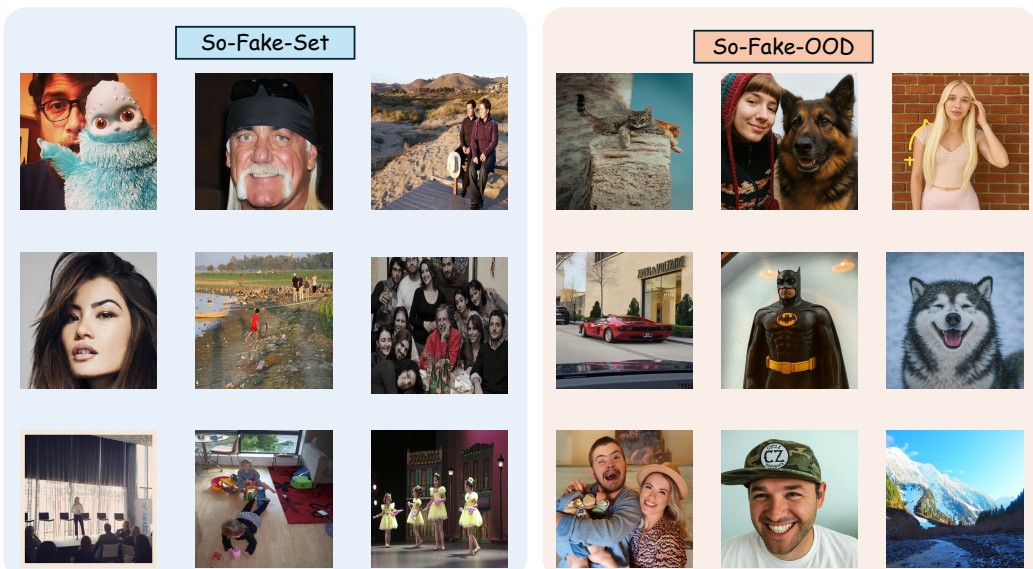

Figure 12: Representative *tampered* images from So-Fake-Set (left) and So-Fake-OOD (right).

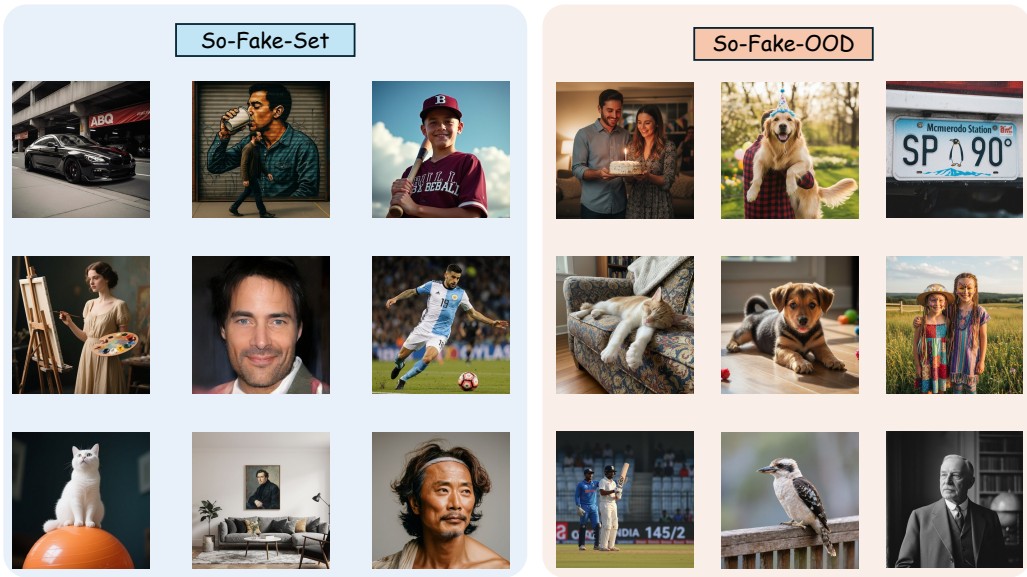

Figure 13: Representative *full synthetic* images from So-Fake-Set (left) and So-Fake-OOD (right).

**Similarity metric:** We compute cosine similarity from DINOv3 embeddings and apply a conservative threshold of 0.9 to identify potential duplicates.

**Results.** The experiment identified 19 high-similarity pairs across the two subsets. We manually inspected all 19 pairs, with representative examples shown in Figure 14. Visual inspection confirms these pairs represent distinct images exhibiting semantic or compositional similarity rather than actual duplication. For instance, Pair 1 shows two photographs of Moraine Lake captured under different lighting and weather conditions, Pair 2 depicts the same vehicle model photographed in different settings, and Pair 5 presents the same species rendered in different color spaces and environmental contexts.

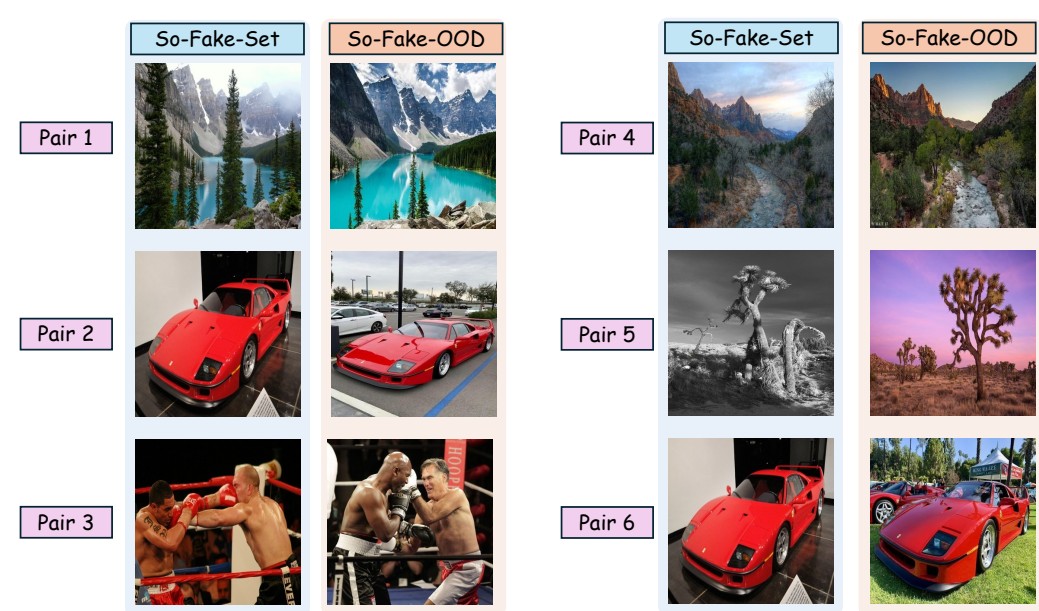

Figure 14: Representative high-similarity pairs identified by DINOv3 (cosine similarity $\geq 0.9$) between So-Fake-Set and So-Fake-OOD. Visual inspection confirms these are distinct images with topic-level similarity rather than actual duplicates, demonstrating minimal overlap between the two splits.

These findings align with expected behavior when comparing large-scale image collections spanning overlapping semantic categories. The minimal overlap rate (19 pairs from 68,300 cross-dataset comparisons, representing 0.028%) confirms that So-Fake-Set and So-Fake-OOD maintain strong distributional independence, validating So-Fake-OOD as a rigorous out-of-distribution benchmark.

## D EXPERIMENTAL SETTINGS AND METHOD DETAILS

In this section, we provide comprehensive implementation details for the So-Fake-R1 framework and experimental configurations used throughout our evaluation. We begin by detailing the reinforcement learning setup, including reward function specifications, training hyperparameters, and optimization procedures (Section D.1). We then outline the implementation specifics for both the baseline methods and our proposed approach, covering model architectures, training protocols, and computational requirements (Section D.2). To further contextualize our contributions, we include a comparative visualization of tampered cases across So-Fake-R1 and competing detectors, highlighting differences in localization quality (Section D.3). Finally, we present additional qualitative examples that illustrate the detection, localization, and explanation capabilities of So-Fake-R1 across diverse forgery types and content categories (Section D.4). These details ensure reproducibility and provide practical guidance for researchers building upon our work.

### D.1 REINFORCEMENT LEARNING CONFIGURATION

As shown in Figure 15, the GRPO training is the core of So-Fake-R1. In this section, we describe the configuration of the GRPO training setup, including the specific weight assignments and detailed reward components. We first discuss the rationale for applying GRPO to multi-modal tasks, then provide detailed specifications of our reward function design and individual reward components.

**GRPO for Multi-modal Tasks.** Reinforcement learning has recently shown promise in vision-language tasks by enabling models to develop reasoning capabilities through trial-and-error learning rather than mimicking human-provided explanations. Among recent advances, Group Relative Policy

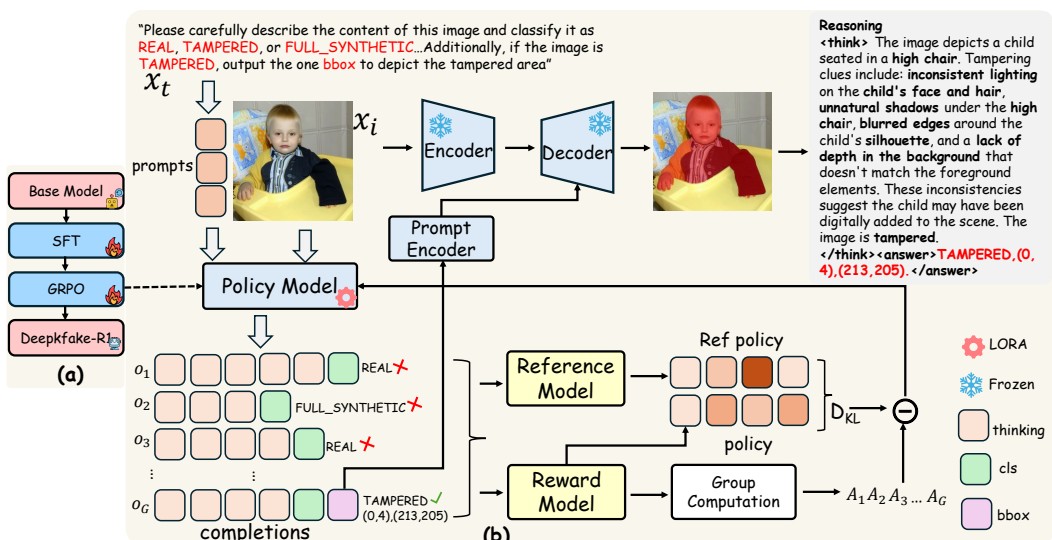

Figure 15: **(a)**: Overview of the So-Fake-R1 training process; **(b)**: The detailed So-Fake-R1 GRPO training process. The example shows a tampered image where a boy has been manipulated.

Table 12: Ablation study on reward weight configurations.

| $\lambda_{\text{fmt}}$ | $\lambda_{\text{cls}}$ | $\lambda_{\text{seg\_fmt}}$ | $\lambda_{\text{IoU}}$ | $\lambda_{\text{L1}}$ | Detection(Acc) | Localization(IoU) |
|---|---|---|---|---|---|---|
| 1.0 | 1.0 | 1.0 | 1.0 | 1.0 | 91.8 | 45.2 |
| 0.1 | 0.9 | 0.1 | 0.9 | 0.9 | **93.2** | **48.6** |
| 0.5 | 1.0 | 0.5 | 1.0 | 1.0 | 92.1 | 46.8 |

Optimization (GRPO) (DeepSeek-AI et al., 2025) has emerged as particularly effective for tasks requiring minimal human supervision, achieving this through rule-based reward mechanisms that guide models toward desired behaviors without relying on extensive annotations. This annotation-free approach makes GRPO particularly valuable for vision-language models (VLMs) (Zhang et al., 2025; Huang et al., 2025a; Xu et al., 2025), where obtaining detailed human explanations for visual reasoning is especially challenging. Notable applications include Seg-Zero (Liu et al., 2025a) for zero-shot segmentation, Visual-RFT (Liu et al., 2025b) for visual question answering, and VLM-R1 (Shen et al., 2025) for robust object detection, establishing GRPO as a standard approach for optimizing multi-modal models that require balancing multiple objectives.

Our reward function design follows established practices in multi-modal reinforcement learning: (1) format rewards ensure structured outputs, which is standard for all structured generation tasks; (2) task-specific rewards based on ground truth labels, following the standard practice of supervised-to-RL conversion; and (3) multi-metric combinations (IoU + L1) that align with established practices in object detection literature. Although GRPO has shown success in various vision-language tasks, it has not yet been applied to forgery detection, where both accuracy and unbiased explainability are essential. To our knowledge, So-Fake-R1 is the first framework to use GRPO for forgery detection.

**Reward Function Design.** The total reward is computed as:

$$R_{\text{total}} = \lambda_{\text{fmt}} R_{\text{fmt}} + \lambda_{\text{cls}} R_{\text{cls}} + \lambda_{\text{seg\_fmt}} R_{\text{seg\_fmt}} + \lambda_{\text{IoU}} R_{\text{IoU}} + \lambda_{\text{L1}} R_{\text{L1}} \quad (2)$$

Based on observations from the ablation study on reward design (Table 7), we found that the progress of GRPO training is less sensitive to formatting rewards, which primarily serve as structural constraints rather than performance drivers. Therefore, we set $\lambda_{\text{fmt}} = \lambda_{\text{seg\_fmt}} = 0.1$, and $\lambda_{\text{cls}} = \lambda_{\text{IOU}} = \lambda_{\text{L1}} = 0.9$, encouraging the model to prioritize classification accuracy and precise segmentation outputs during training. This weight allocation is empirically validated through Table 12, demonstrating that each component contributes to the final performance. Below, we detail each reward component:

*Explanation Format Reward ($R_{fmt}$).* This reward function encourages structured reasoning by requiring the model to format its output using `<think>...</think>` for the reasoning process and `<answer>...</answer>` for the final answer. The model receives a reward of +1 if the output follows this format correctly; otherwise, it receives 0. This follows the standard practice in structured text generation tasks where format compliance is enforced through binary rewards.

*Detection Reward ($R_{cls}$).* This reward encourages accurate multi-class classification among `REAL`, `TAMPERED`, and `FULL_SYNTHETIC`, based on the label provided within the `<answer>...</answer>` tags. The model receives a reward of +1 for correctly identifying `REAL` or `FULL_SYNTHETIC` images, and a higher reward of +3 for correctly identifying `TAMPERED`, as detecting tampered images is more challenging based on our preliminary analysis showing lower baseline performance on this class. Incorrect classifications receive a reward of 0.

*Localization Format Reward ($R_{seg\_fmt}$).* This reward ensures that bounding boxes follow a strict coordinate specification. The model receives a reward of +1 if its output includes properly formatted coordinates (i.e., four numerical values enclosed between the tags `<|box_start|>` and `<|box_end|>`), such as: `<|box_start|>` $(x_1, y_1), (x_2, y_2)$ `<|box_end|>`. If the format is incorrect or missing, the reward is 0.

*IoU Reward ($R_{IoU}$).* This reward grants a positive score when predicted boxes achieve meaningful overlap with ground truth. For `TAMPERED` images, the model receives a reward of +1 if the predicted bounding box achieves an Intersection-over-Union (IoU) greater than 0.5 compared to the ground truth. The 0.5 IoU threshold follows the standard practice in object detection literature for meaningful overlap assessment. For `REAL` and `FULL_SYNTHETIC` images, which do not require localization, a reward of +1 is assigned by default. In all other cases, the reward is 0.

*L1 Reward ($R_{L1}$).* This reward further refines localization accuracy by rewarding predictions within close proximity to ground truth coordinates. For `TAMPERED` images, the model receives a reward of +1 if the total L1 distance across all four coordinates is less than 10 pixels. For `REAL` and `FULL_SYNTHETIC` samples, which do not require bounding boxes, a default reward of +1 is assigned. Otherwise, the reward is 0. The combination of IoU and L1 rewards ensures both region overlap quality and precise boundary alignment, as IoU alone can be satisfied by oversized boxes.

## D.2 IMPLEMENTATION DETAILS

**Baseline Methods.** For detection-only methods including CnnSpot (Frank & Holz, 2021), Uni-vFD (Ojha et al., 2023), FreAware (Tan et al., 2024b), and NPR (Tan et al., 2024a), we follow the official implementation guidelines provided in their respective documentation and adopt the recommended or highest-performing configurations when available.

For image forgery detection and localization (IFDL) methods, we fine-tune TruFor (Guillaro et al., 2023), PSCC-Net (Liu et al., 2022), and SIDA (Huang et al., 2025b) on So-Fake-Set according to the official recommended settings. For FakeShield (Xu et al., 2024) and HIFI-Net (Guo et al., 2024), we use the pre-trained weights for evaluation due to code availability constraints.

For vision-language models, including LLaVA-1.5-13B (Liu et al., 2023), InternVL3-8B (Zhu et al., 2025), Qwen2.5-VL-7B (Bai et al., 2025), and DeepSeek-VL-7B (DeepSeek-AI et al., 2025), we adopt the `ms-swift`[5] framework for streamlined integration, fast inference, and effective hyperparameter tuning. For each model, we select the best-performing checkpoint based on validation performance. For LISA (Lai et al., 2024), we use the official codebase and follow the authors' recommended hyperparameter settings.

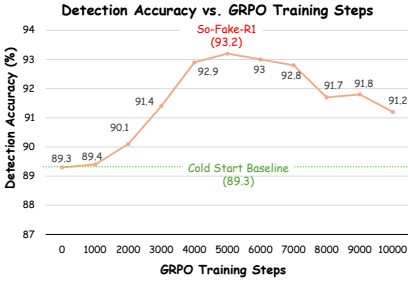

Figure 16: Detection accuracy over GRPO training steps.

**So-Fake-R1 Implementation.** We use Qwen2.5-VL-7B-Instruct Bai et al. (2025) as our policy model and SAM2 Ravi et al. (2024) for segmentation refinement. Our training follows a two-stage pipeline comprising a cold-start phase followed by GRPO fine-tuning. In both stages, all input images (and

---

[5]`https://github.com/modelscope/ms-swift`

masks, when present) are resized to $224 \times 224$ pixels to ensure consistent input dimensions and reduce memory consumption.

For the cold-start phase, we apply LoRA (Hu et al., 2021) with $\alpha = 32$ and rank 16. The model is trained using a learning rate of 5e-5, weight decay of 0.1, and a maximum token length of 2048. This stage takes approximately 30 minutes to complete on a single A100 GPU (40GB).

For the GRPO phase, we also apply LoRA with $\alpha = 32$ and rank 8. The model is trained with a learning rate of 1e-4, a warmup ratio of 0.05, weight decay of 0.1, and a maximum token length of 2048. We assign reward weights as $\lambda_{\text{fmt}} = \lambda_{\text{seg\_fmt}} = 0.1$, and $\lambda_{\text{cls}} = \lambda_{\text{IOU}} = \lambda_{\text{L1}} = 0.9$ based on Table 12. GRPO training is conducted on two A100 GPUs (40GB each) and completes in approximately 24 hours. We select the checkpoint at 5000 training steps, as it achieves the best overall performance across evaluation metrics, as shown in Figure 16.

### D.3 QUALITATIVE COMPARISON ON TAMPERED CASES

We further provide a qualitative comparison of tampered cases against representative IFDL baselines. As shown in Figure 17, So-Fake-R1 achieves more precise localization of manipulated regions, closely matching the ground-truth masks. In contrast, competing methods often misidentify boundaries or overlook subtle edited areas. These results highlight the effectiveness of our reinforcement learning–based framework in capturing fine-grained tampering artifacts.

### D.4 ADDITIONAL QUALITATIVE EXAMPLES

In this section, we present additional examples of So-Fake-R1's outputs on the So-Fake-OOD benchmark. Since So-Fake-R1 was not trained on this benchmark, the results include both successful predictions and failure cases. To ensure fair representation, test images were randomly selected. These examples illustrate the model's generalization capabilities, highlight areas for improvement, and suggest directions for future research. The qualitative examples are shown in Figures 18–25.

## E BROADER SOCIAL IMPACT

To ensure quality, fairness, and responsible use, we incorporated multiple safeguards throughout the development process. Expert reviewers were engaged at every stage, including the selection of source content, the validation of generative outputs, and the refinement of textual explanations, to guarantee both reliability and appropriateness. For So-Fake-OOD, we followed Reddit's Public Content Policy and applied multi-stage human filtering to exclude unsuitable or sensitive material. All datasets and models are released strictly for non-commercial, research-only purposes under controlled access.

So-Fake-Set, So-Fake-OOD, and So-Fake-R1 framework are designed to advance the field of multi-modal forgery detection, with a particular focus on social media contexts. By offering large-scale, diverse, and well-annotated benchmarks alongside an interpretable and high-performing model, our work provides valuable resources for the research community. These contributions can support future developments in robust AI systems, foster academic exploration, and assist in building more trustworthy digital media ecosystems. We believe our dataset and methods will positively impact the broader AI and computer vision communities by encouraging progress in transparent, explainable, and socially beneficial technologies for image authenticity verification.

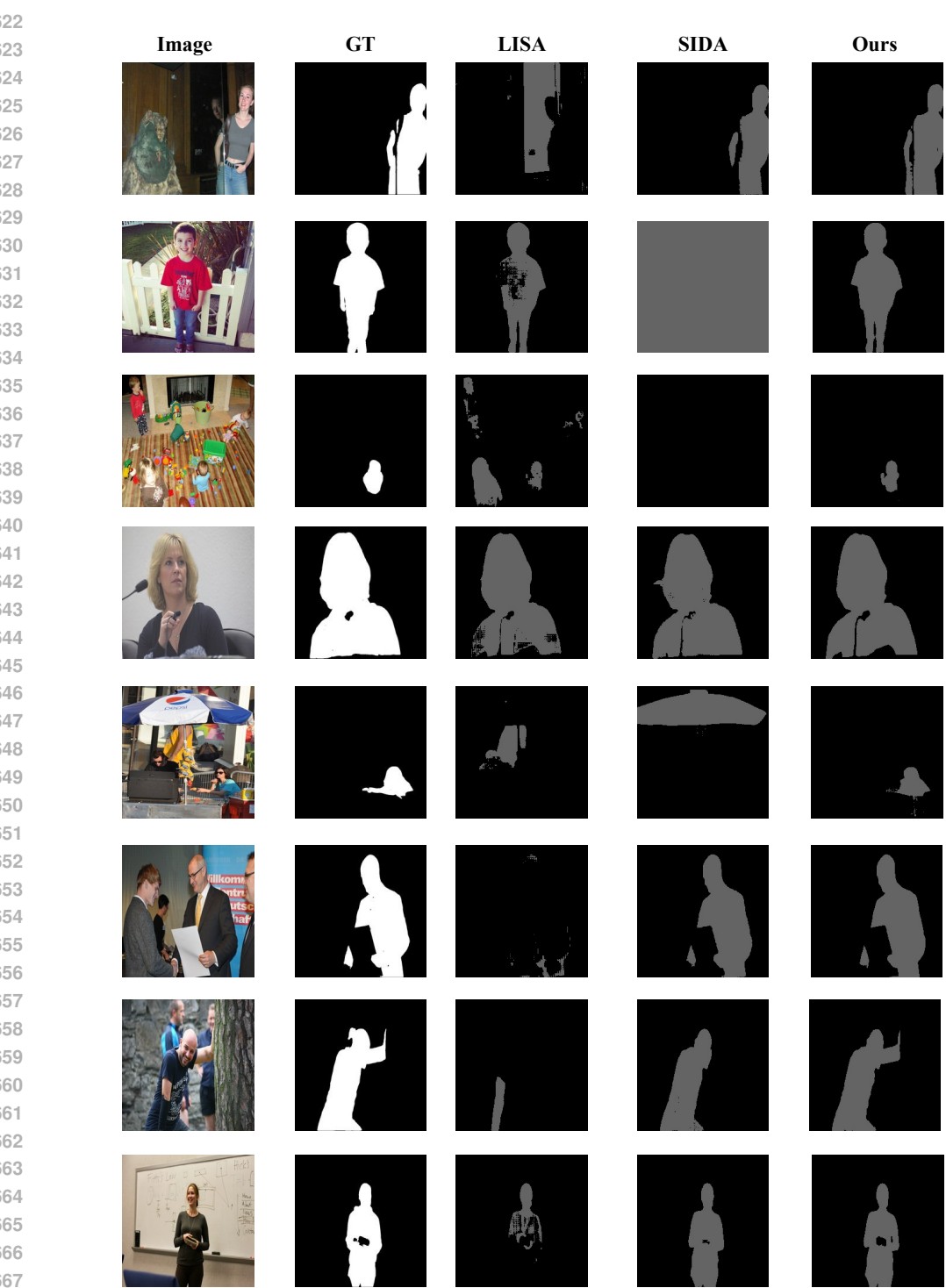

Figure 17: Qualitative comparison of tampered cases on **So-Fake-Set**, comparing ground-truth (GT), LISA, SIDA, and our So-Fake-R1.

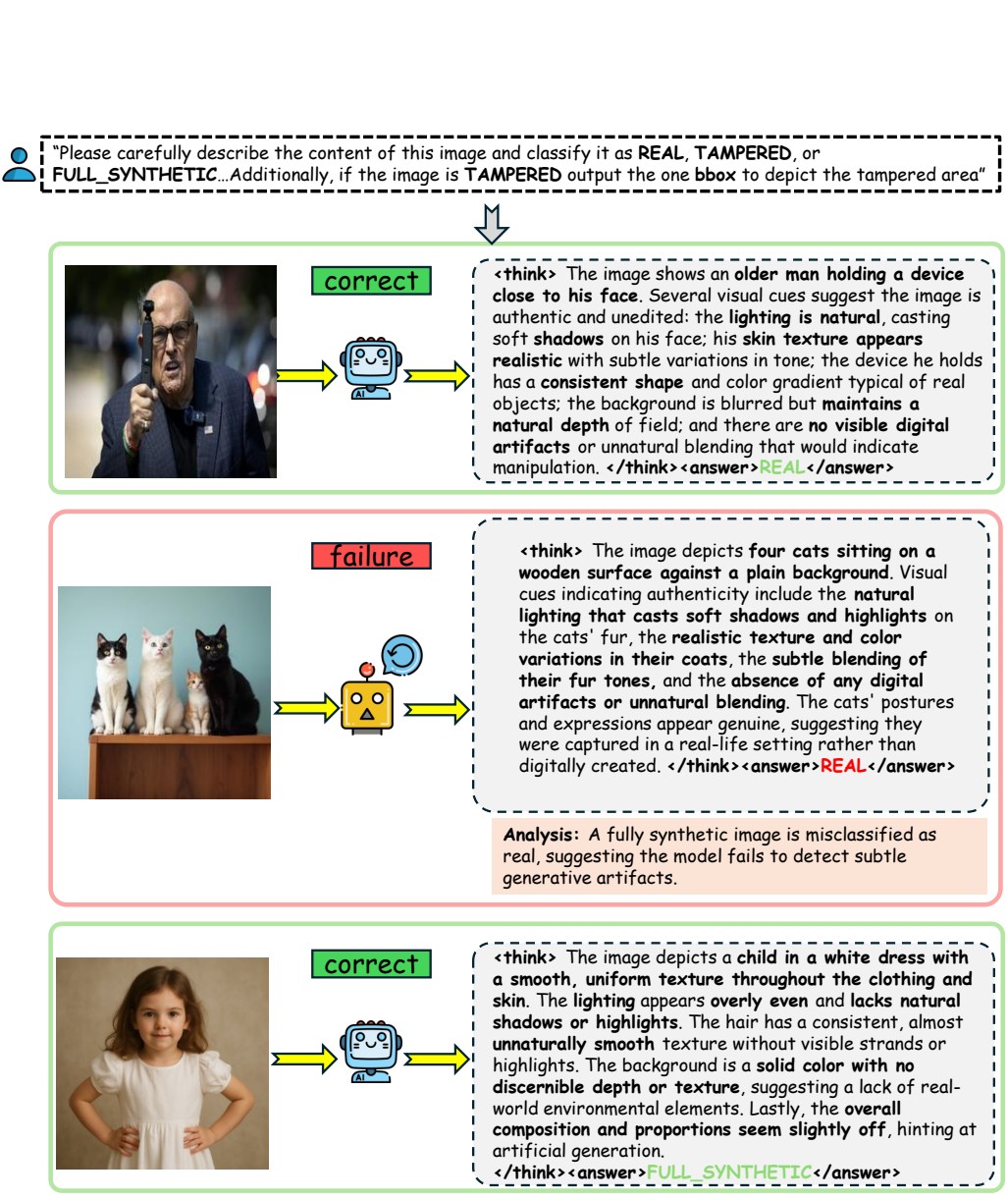

Figure 18: Qualitative results of So-Fake-R1 on the So-Fake-OOD benchmark.

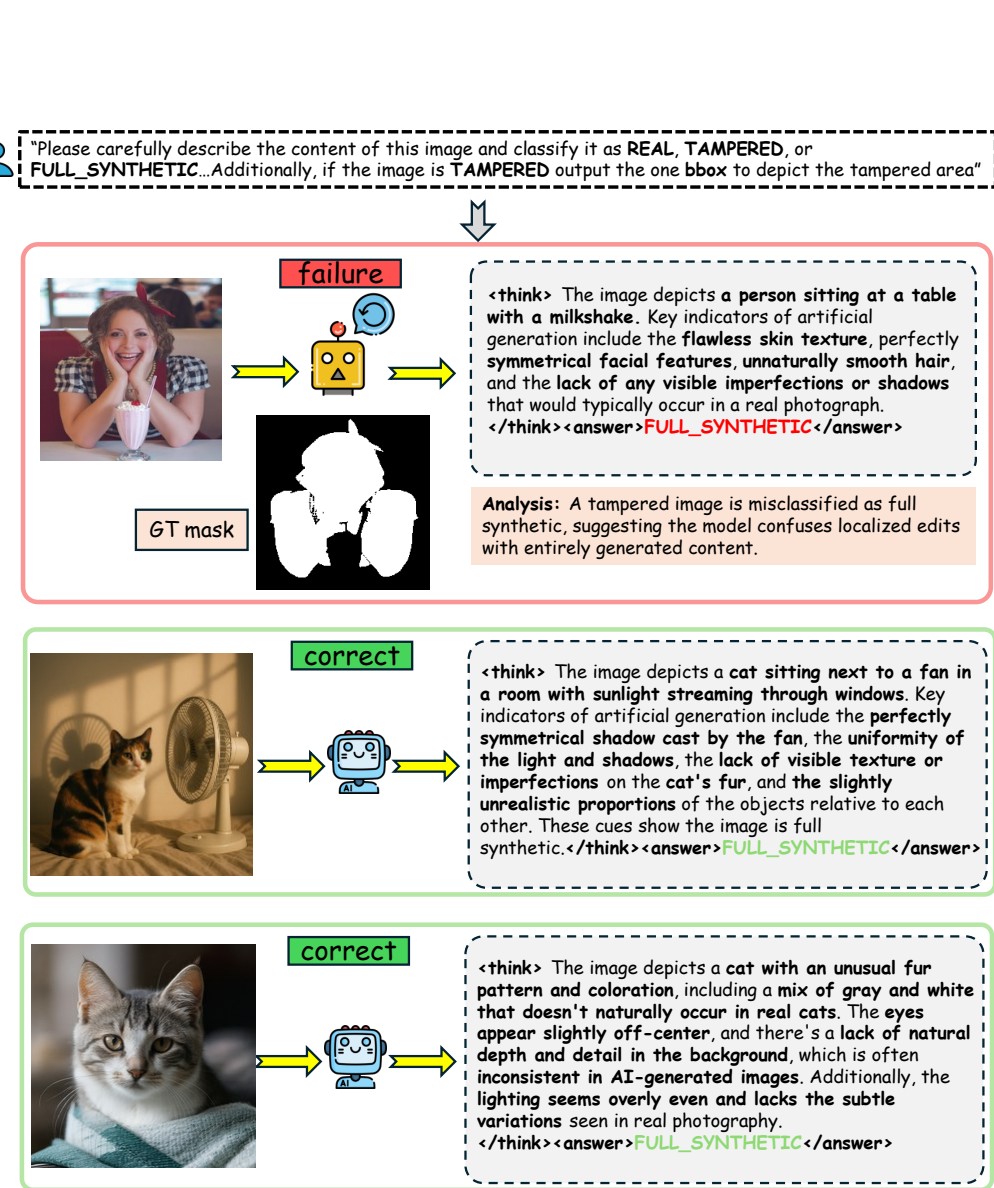

Figure 19: Qualitative results of So-Fake-R1 on the So-Fake-OOD benchmark.

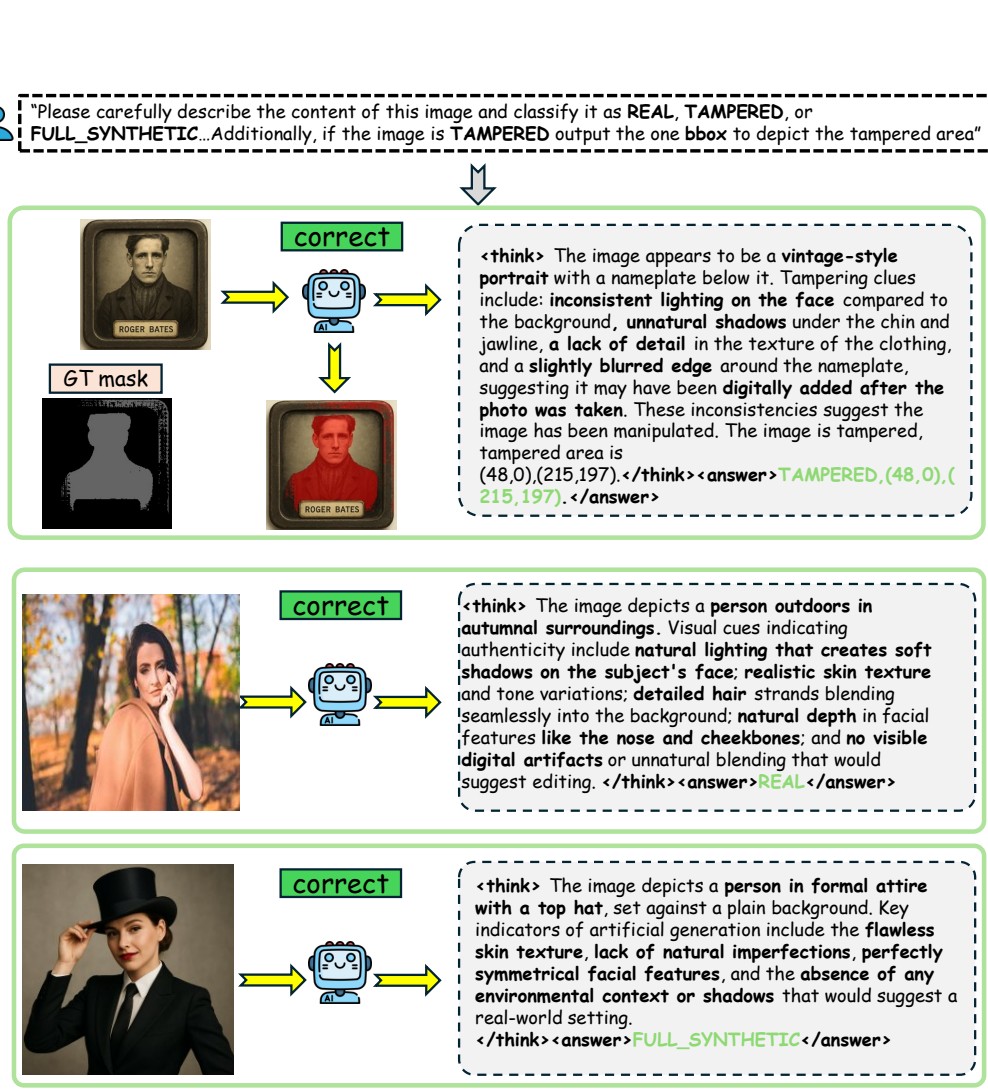

Figure 20: Qualitative results of So-Fake-R1 on the So-Fake-OOD benchmark.

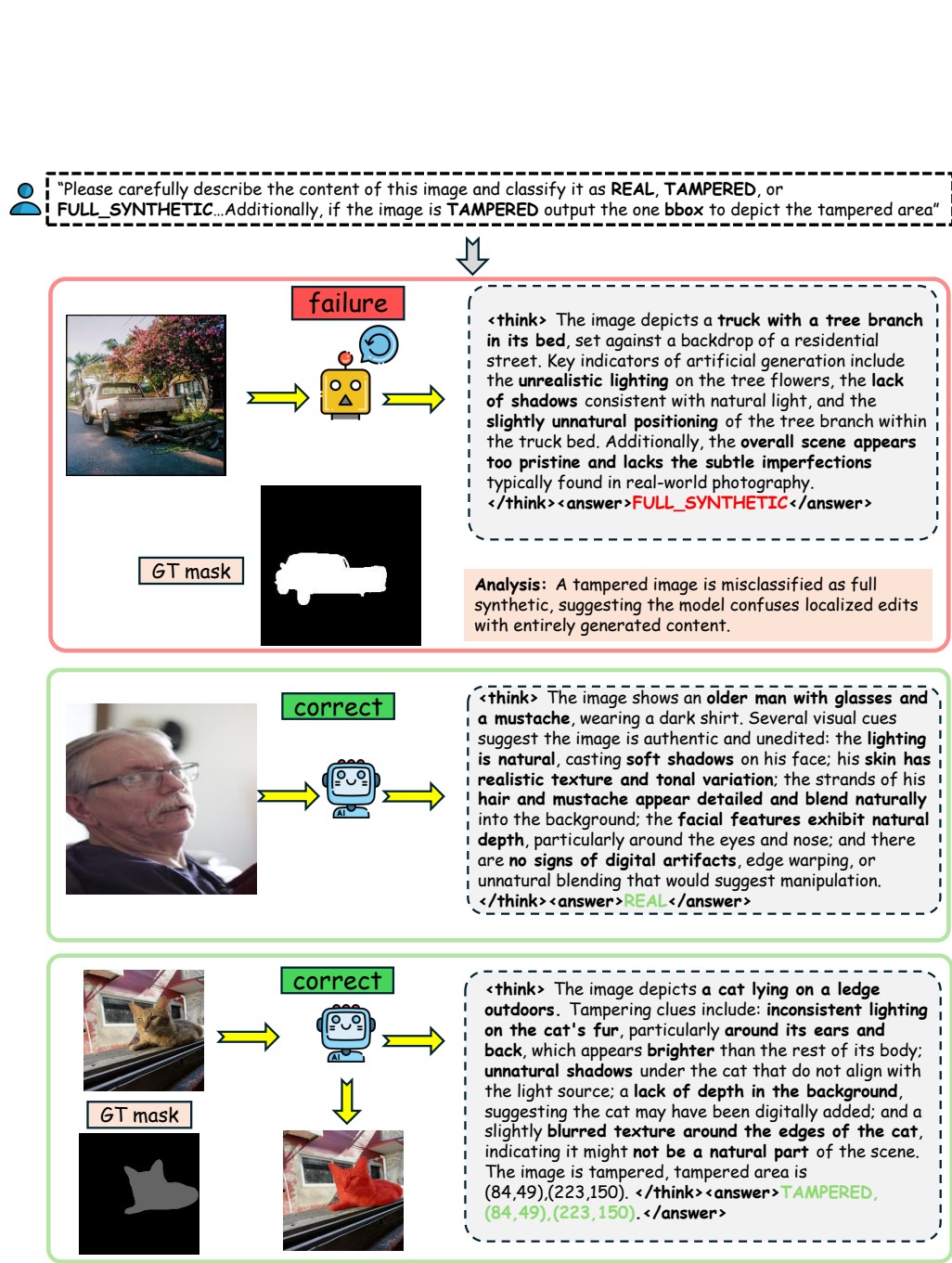

Figure 21: Qualitative results of So-Fake-R1 on the So-Fake-OOD benchmark.

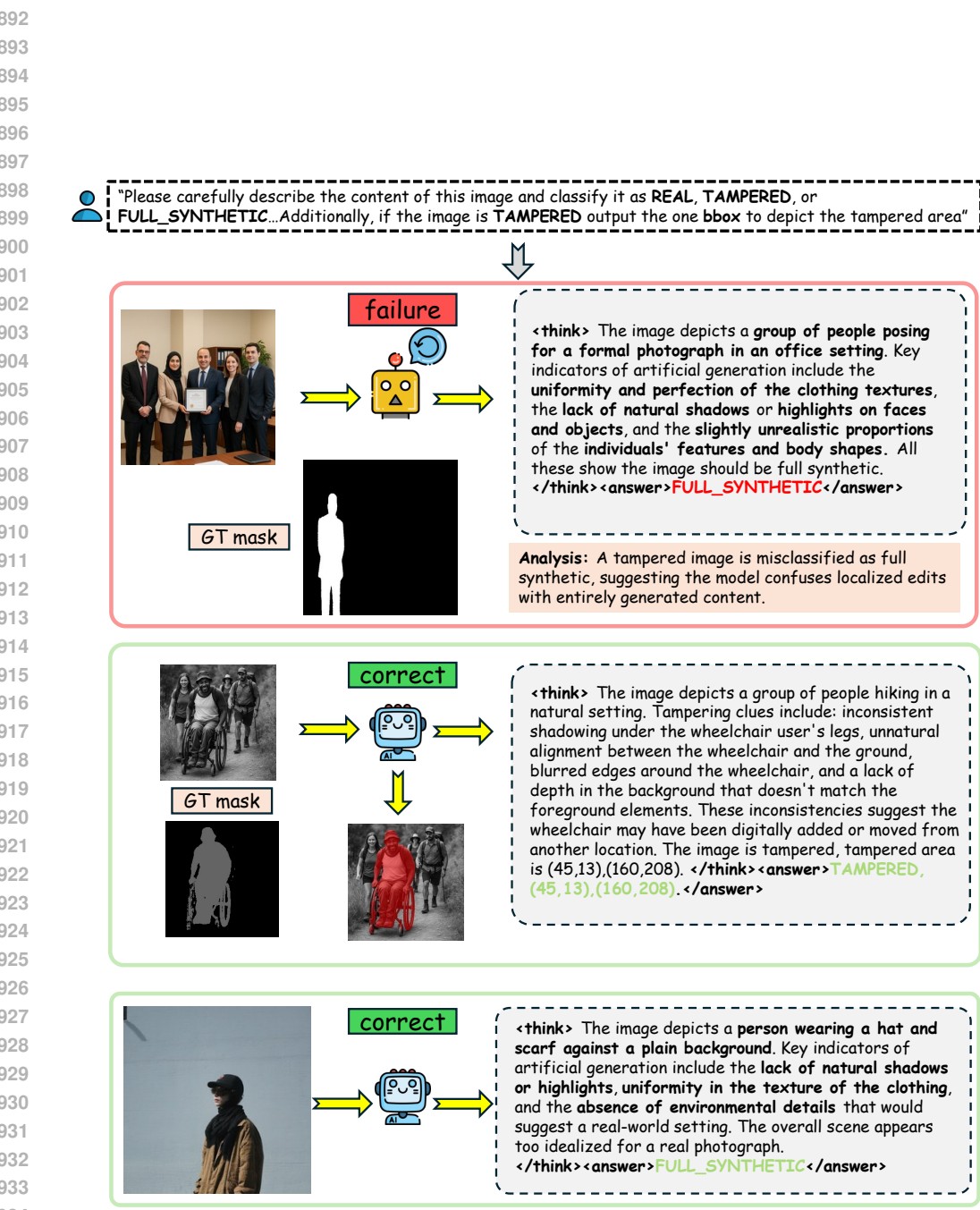

Figure 22: Qualitative results of So-Fake-R1 on the So-Fake-OOD benchmark.

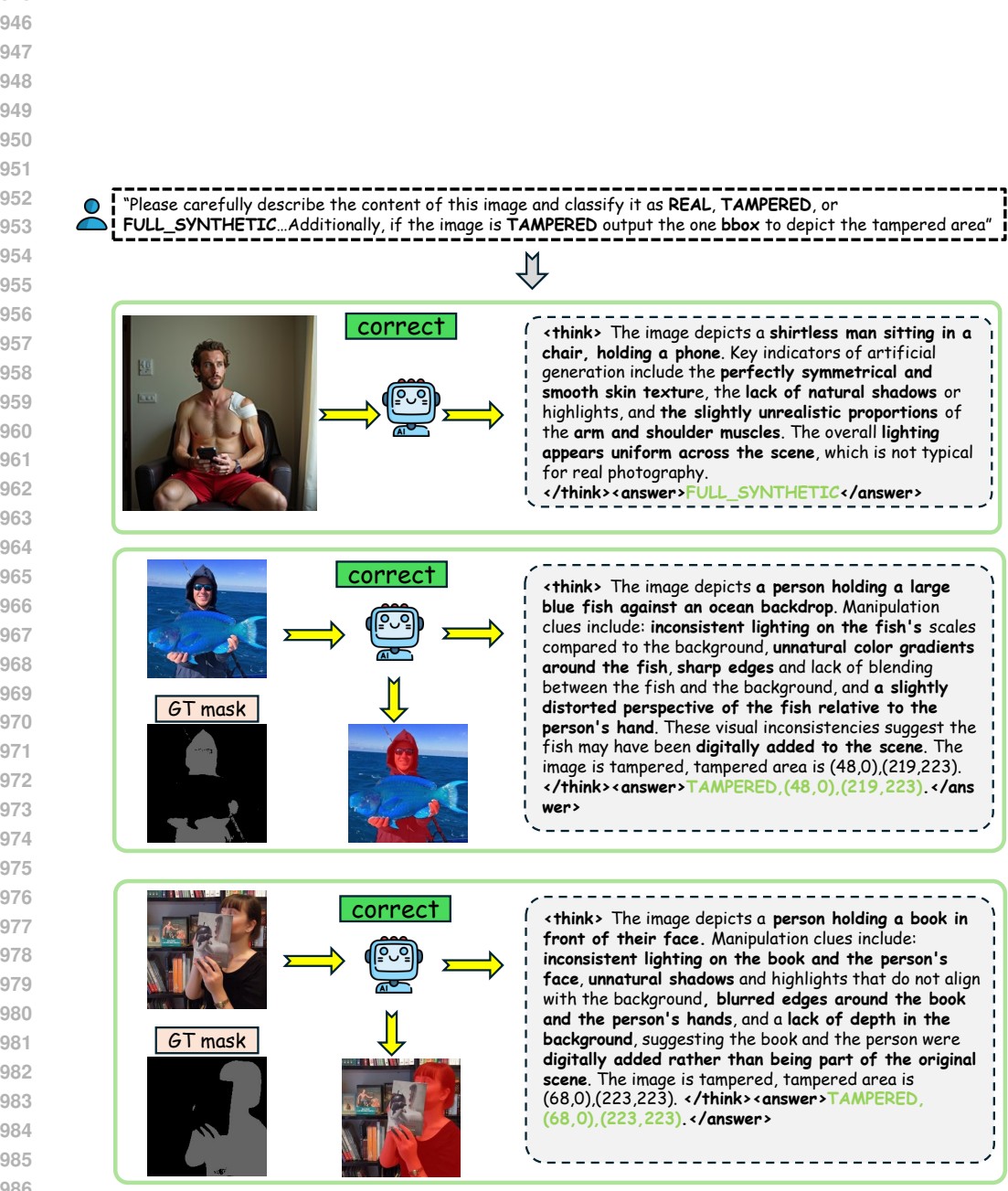

Figure 23: Qualitative results of So-Fake-R1 on the So-Fake-OOD benchmark.

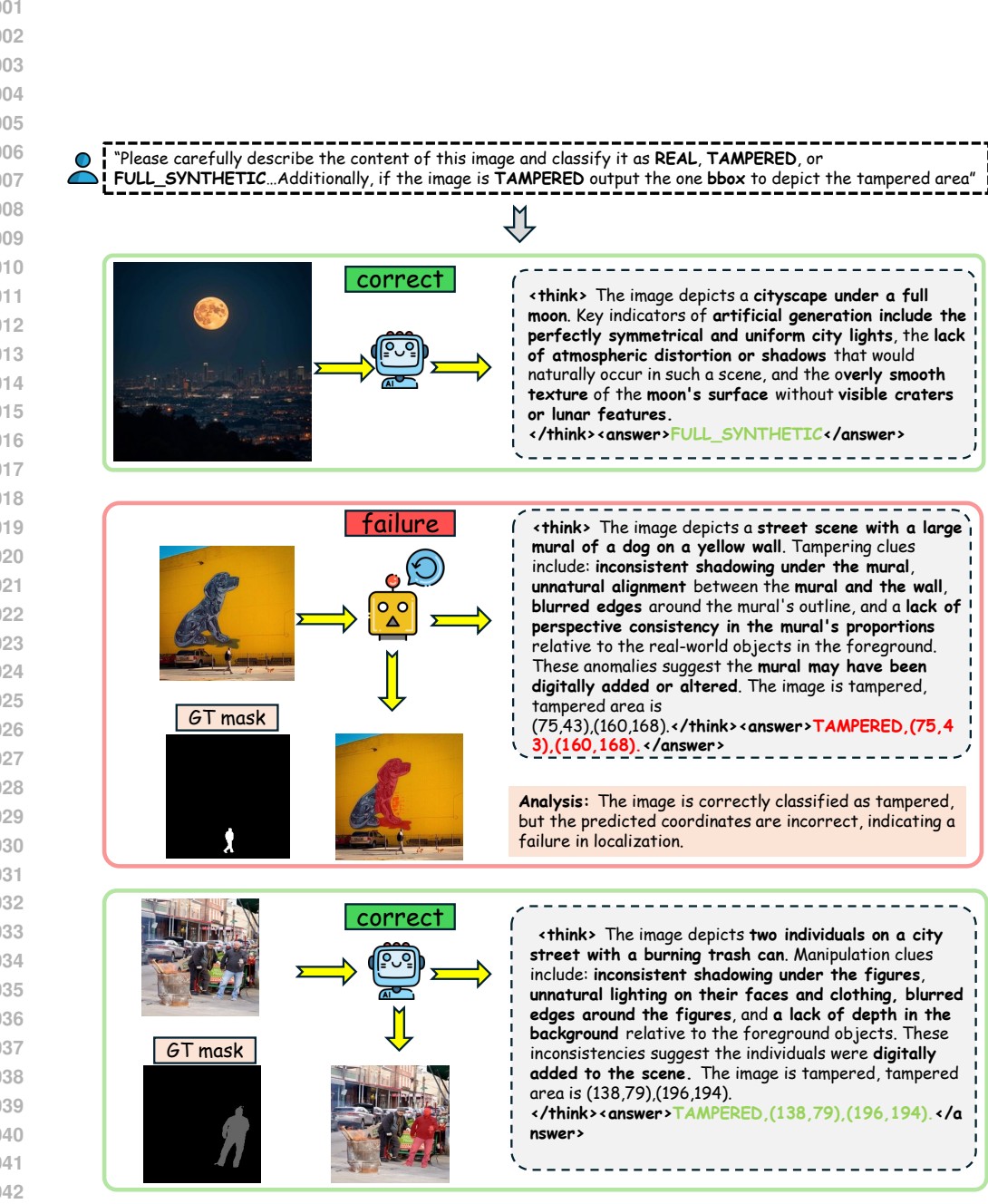

Figure 24: Qualitative results of So-Fake-R1 on the So-Fake-OOD benchmark.

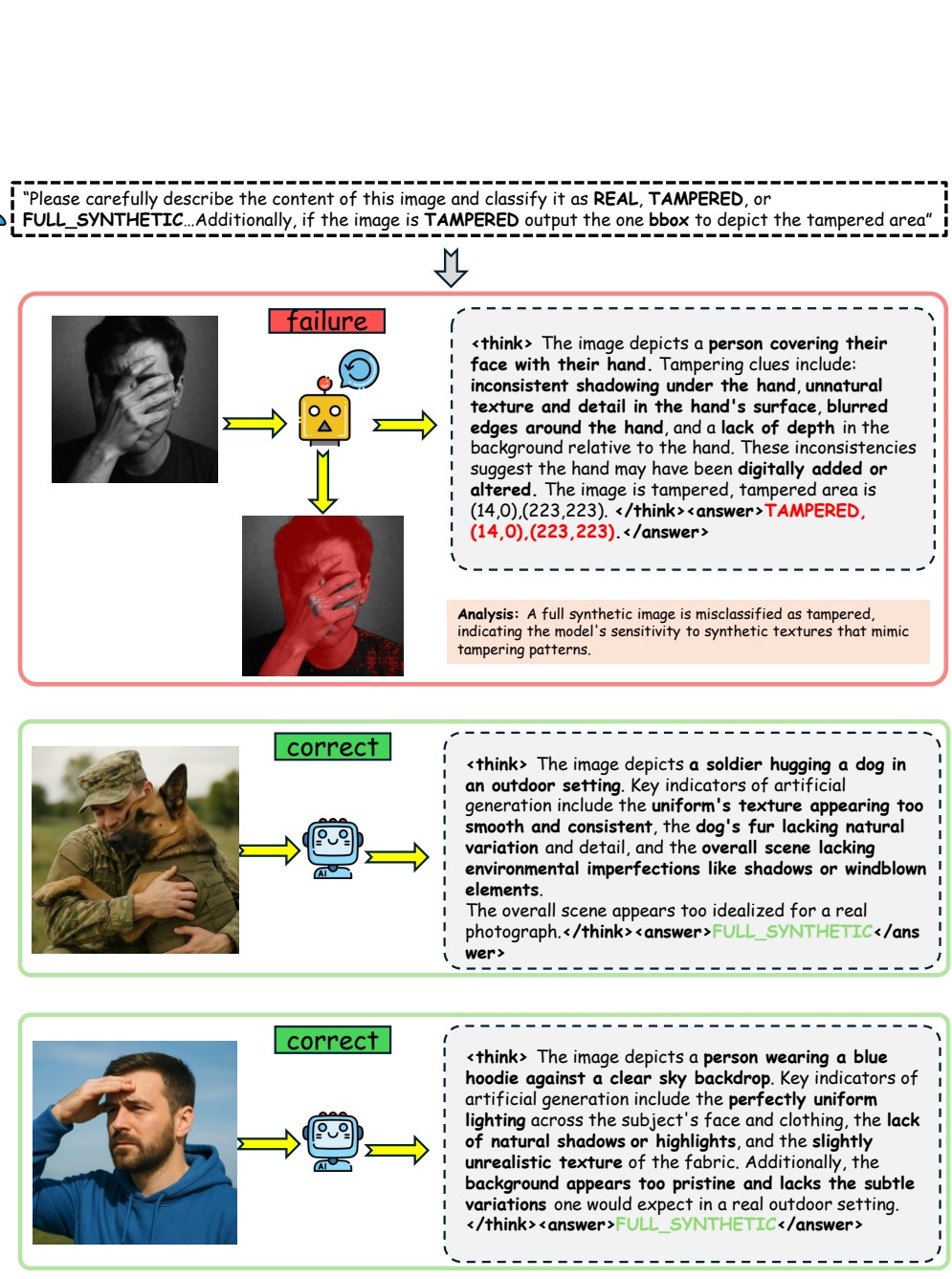

Figure 25: Qualitative results of So-Fake-R1 on the So-Fake-OOD benchmark.

