# OpenReview forum: "So-Fake: Benchmarking Social Media Image Forgery Detection"
_ICLR.cc/2026/Conference — Submitted to ICLR 2026_

### Official Review · Reviewer_npdH · 2025-10-25

**Soundness:** 3
**Presentation:** 2
**Contribution:** 3
**Rating:** 6
**Confidence:** 4

**Summary:**

This paper introduces So-Fake, a large-scale benchmark aimed at evaluating forgery detection on social media imagery. It includes two core datasets — So-Fake-Set (2M images for training/validation) and So-Fake-OOD (100K out-of-distribution samples) — designed to simulate real-world generative diversity using varied GAN- and diffusion-based models. The accompanying model, So-Fake-R1, integrates reinforcement learning to jointly optimize detection, localization, and explanation tasks. Experiments demonstrate So-Fake’s realism, diversity, and diagnostic value for understanding model generalization.

**Strengths:**

1. Comprehensive dataset design: The paper presents a well-structured, large-scale benchmark that captures both in-domain and cross-domain conditions, using authentic social media content and realistic generation pipelines.
2. Unified evaluation and interpretability: The incorporation of reinforcement learning (GRPO) for multi-objective optimization across detection, localization, and explanation is technically sound and represents a meaningful advancement over prior benchmarks that treat these tasks independently.
3. Authentic OOD evaluation: So-Fake-OOD leverages real Reddit data paired with commercial generative models entirely disjoint from training, creating a realistic testbed that mirrors actual deployment scenarios with genuine distributional shifts.
4. Extensive ablation demonstrate the effectiveness of the proposed training recipe of So-Fake-R1.

**Weaknesses:**

1. Insufficient ablation on scalability: The paper lacks detailed analysis on how dataset size or class diversity impacts generalization performance, which would strengthen claims of robustness.
2. Potential domain limitations: While Reddit imagery provides diversity, reliance on a single platform may bias the dataset’s domain distribution and restrict its representation of global social media conditions.

**Questions:**

See weakness.

---

> ### Author Response · Authors · 2025-11-19
> **Response to Reviewer npdH**
>
> **W1: Insufficient ablation on scalability**
>
> Thank you for this valuable suggestion. We agree that understanding scalability is essential for demonstrating robustness.
>
> **We have conducted data scale experiments.** Specifically, we trained CNNSpot on 10%, 30%, 50%, and 100% of So-Fake-Set and evaluated on So-Fake-OOD. When sampling subsets, we ensured coverage of all 30 generative methods and maintained balanced proportions across the three classes (real, tampered, fully synthetic), guaranteeing that each subset reflects the full distribution of manipulation types and generators. As shown in Table 1, the results show consistent performance gains as data volume increases: detection accuracy improves from 56.3% (10%) to 65.2% (100%), confirming that larger training sets enhance cross-domain generalization while preserving diversity.
>
> **Table 1: CNNSpot OOD accuracy vs. training size.**
> | Training Portion | Detection Acc (%) |
> |------------------|-------------------|
> | 10%              | 56.3              |
> | 30%              | 57.4              |
> | 50%              | 60.8              |
> | 100%             | 65.2              |
>
> **Regarding class diversity, we provide extensive analysis in Appendix C.3.** First, we analyze task-type diversity by comparing tampered vs. fully synthetic forgeries (Figure 9), showing that models trained on a single manipulation type generalize poorly to the other (0.578 AUROC), while our unified three-class approach maintains robust performance across both (0.767 and 0.724 AUROC). Second, we examine the impact of content diversity through face vs. non-face experiments (Figure 10), demonstrating that models trained exclusively on facial content suffer a 39.8% performance drop on non-face categories, while our 12-class taxonomy yields balanced performance across diverse content types.
>
> We acknowledge that more fine-grained ablations on category-level scaling would further strengthen our analysis. We plan to conduct systematic experiments varying the number of semantic categories and manipulation methods in future work.
>
> **W2: Potential domain limitations**
>
> Thank you for raising this important point about platform diversity. We agree that no single platform can fully represent the global social-media ecosystem.
>
> **Our choice of Reddit is driven by practical and ethical considerations.** Many mainstream platforms (Instagram, Facebook, TikTok, Weibo) prohibit large-scale academic data collection due to privacy and copyright restrictions, making them unsuitable for constructing a legally compliant research dataset. Reddit is among the few major platforms that permits academic-scale image collection under clearly defined terms of service, allowing us to build an open and reproducible benchmark.
>
>
> **Despite originating from a single platform, So-Fake-OOD captures substantial diversity characteristic of social media content.** Reddit images exhibit varied resolutions, multi-stage compression, reposting artifacts, and diverse camera sources. Our 12-category taxonomy ensures broad topical coverage, ranging from portraits and landscapes to objects and events. This design allows So-Fake-OOD to reproduce the distributional characteristics that detectors encounter in real-world deployments: authentic user-generated content mixed with synthetic imagery from commercial generators.
>
> **We acknowledge this as a limitation of the current benchmark.** Cross-platform generalization remains an important research direction as data-access policies and ethical frameworks evolve to permit broader academic collection.
>
> We thank the reviewer for this thoughtful observation and hope our response clarifies the rationale behind our platform selection.

---

> ### Comment · Area_Chair_GMV5 · 2025-11-28
> **Rebuttal Review Request**
>
> Dear Reviewers,
>
> Thank you for your time and thoughtful feedback on this manuscript.
>
> The authors have now submitted their rebuttal. If you haven’t already, we kindly ask you to review their responses and consider whether your concerns have been adequately addressed.
>
> Best regards,
>
> AC

---

### Official Review · Reviewer_GQQN · 2025-10-27

**Soundness:** 2
**Presentation:** 3
**Contribution:** 2
**Rating:** 4
**Confidence:** 5

**Summary:**

This paper proposes So-Fake, which includes two datasets (So-Fake-Set and So-Fake-OOD) and a detector named So-Fake-R1. From the dataset perspective, it covers multiple data sources, 12 semantic classes, and adopts various generation methods for synthesis. Additionally, the OOD dataset collects a large number of images from social media platforms (e.g., Reddit) to simulate real-world scenarios. From the detector perspective, a reinforcement learning-based joint detector is proposed to simultaneously handle three tasks: detection, localization, and explanation. Experimental results show that the proposed method outperforms some existing works under the condition of using the same training set.

**Strengths:**

- Proposes a new dataset that covers richer semantic categories, synthesis methods, and real-world social media data.
- Presents a reinforcement learning-based detector with aligned rewards designed for the three tasks of detection, localization, and explanation.
- Conducts sufficient comparative experiments and ablation studies to verify the effectiveness of the proposed detector.

**Weaknesses:**

Dataset-related Issues:
- The paper claims that existing datasets have a "Narrow Categorical Scope", yet the proposed dataset does not significantly expand the semantic categories (only 12 categories as shown in Figure 2). Compared with existing datasets, what types of data are completely newly included? Furthermore, for the AIGC Detection task, is there a genuine need for such a large number of semantic categories? Does covering more semantic categories truly contribute to improving generalization performance?
- Regarding the claim of addressing "Outdated Generation Quality", similarly, is it really necessary to cover so many synthesis methods to achieve good out-of-domain (OOD) detection performance? Could a training set composed of specific combinations (e.g., StyleGAN3 + Latent Diff) yield unexpected OOD performance? Although the paper invests significant effort in creating new data using various synthesis algorithms, it fails to analyze whether these data are beneficial for building a detector with stronger generalization. Exploring the dataset from this perspective and deriving conclusions would better address future generation algorithms.
- Although many social media images are collected as OOD and real-scenario data, what are the differences between these data and other datasets? If they are more difficult to detect, what are the underlying reasons? Is it due to unique traces introduced by processing on social platforms? These questions are worth investigating.
- In Section 3.3 "Data Generation", the data generation approach seems similar to that of previous datasets. Are there any unique innovations? Otherwise, the mere application of different generation algorithms makes this section more like an experimental report rather than an academic research contribution.

Method:
- The techniques involved, such as cold start and GRPO training, are all based on existing works. Although there may be certain improvements in designing feedback rewards for the three tasks (detection, localization, and explanation) simultaneously, the core originality is insufficient.

Experimental Limitations
- All experiments in this paper are based on the proposed So-Fake-Set. Although the paper states that comparative methods are fine-tuned on this dataset, it cannot guarantee that the performance achieved through these fine-tunings is optimal. Adding cross-dataset experiments would better verify the effectiveness of the proposed method. For example, training the proposed So-Fake-R1 on the dataset introduced by CAT-Net (IJCV'22), then comparing it with methods like CAT-Net and TruFor on other OOD datasets.

Justification: Although the paper invests substantial effort in constructing the dataset, it fails to conduct effective information mining on the constructed data to derive innovative conclusions. While this dataset may serve as a baseline to inspire future work, it is difficult for me to give it a highly positive rating at this stage.

**Questions:**

Please see Weaknesses.

---

> ### Author Response · Authors · 2025-11-19
> **Response to Reviewer GQQN: W1**
>
> W1: Thank you for raising this important question. We address it in three parts: **(Q1) What types of data are newly included, and what is their purpose?**, **(Q2) Is there a genuine need for such a large number of semantic categories?** and **(Q3) whether semantic diversity genuinely benefits** **AIGC** **detection generalization**.
>
> **Q1: What types of data are newly included, and what is their purpose?**
>
> **Social Media-Grounded Category Design.** Our aim is not to introduce arbitrary or artificially “new” semantic classes, but to ensure that the dataset reflects **the actual content distribution of social platforms**. To this end, we analyzed over 100K authentic Reddit images and derived the 12 categories that genuinely dominate social media and are commonly subject to manipulation. These categories are newly included relative to existing forgery datasets not because they are novel in isolation, but because they capture real social-media posting and manipulation patterns that prior benchmarks do not cover.
>
>  **Expanded Content Coverage Beyond Existing Datasets.** Most existing forgery datasets focus narrowly on faces (e.g., FaceForensics++, DFFD) or cover only a single semantic domain, and many do not explicitly specify which semantic categories. In contrast, So-Fake spans all 12 content categories observed in real social media. This broader coverage serves two purposes: (1) it mirrors the heterogeneous nature of user-generated content, and (2) it enables evaluating detectors across diverse semantic contexts—an essential requirement for real-world deployment, where models encounter a wide range of scenes and subjects.
>
> **Q2: Is there a genuine need for such a large number of semantic categories?**
>
>  In a social media–oriented benchmark, the role of semantic breadth is not to increase classification granularity but to prevent detectors from overfitting to category-specific artifacts. Detectors trained on narrow domains (e.g., only faces) often rely on spurious cues tied to a limited set of content types and fail when encountering unfamiliar scenes. In contrast, real social media contains highly diverse contexts, objects, and visual styles. A broader set of categories is therefore necessary to construct evaluation protocols that mirror realistic deployment conditions. As shown in the Table 1, semantic diversity directly contributes to cross-category robustness, highlighting that such breadth is a practical requirement.
>
> **Q3: Whether semantic diversity genuinely benefits AIGC detection generalization?**
>
> We have conducted experiments to evaluate this question (Section C.3, Figure 10). To ensure a fair comparison, we kept the training data size identical across settings and varied only the semantic scope. Using CNNSpot as the detector, we compared models trained on different semantic scopes:
>
> **Table 1: Controlled Experiment: Face vs. Non-Face Generalization**
> | Training Setting     | Face AUROC | Non-Face AUROC |
> |----------------------|------------|-----------------|
> | Face-only training   | 0.960      | 0.562           |
> | Multi-category       | 0.897      | 0.848           |
>
> This controlled experiment shows that a detector trained only on faces performs well on face content but collapses on non-face content (a drop from 0.960 to 0.562). In contrast, multi-category training yields balanced performance across both domains and improves non-face detection substantially (0.562 → 0.848). These results demonstrate that semantic diversity is essential for preventing overfitting to category-specific artifacts and for achieving robust cross-category generalization. This validates our decision to include 12 categories rather than focusing on a single content type.
>
> The face vs. non-face comparison provides the most diagnostic signal for evaluating semantic diversity, as it represents the clearest contrast between the historically dominant content type in forgery datasets (faces) and the broader visual landscape of social media. These results conclusively demonstrate that semantic breadth is essential for cross-category robustness. A comprehensive ablation across all 12 individual categories would offer additional granularity and remains a valuable direction for future work.
>
>  We thank the reviewer for this question and hope our response addresses this concern.

---

> ### Author Response · Authors · 2025-11-19
> **Response to Reviewer GQQN: W2**
>
> Thank you for this important question. To clarify the scope and necessity of our synthesis-method diversity, we address the reviewer’s concerns in three parts: **(Q1) Is extensive synthesis method coverage necessary?** **(Q2) Whether a small subset of generators could suffice?** and **(Q3) Whether diverse generation methods concretely improve generalization?**
>
> **Q1: Is extensive synthesis method coverage necessary?**
>
> We fully agree with the reviewer that establishing robust cross-generator generalization is a core challenge in image forensics. In many binary detection settings, a few representative generators might suffice. However, our empirical analysis reveals that the social media context introduces unique complexities that demand broader synthesis coverage. Our goal is not to exhaustively include every possible generator, but to support a **three-way social-media forgery setting (real / tampered / full-synthetic)**, which fundamentally differs from cross-generator detection. In real social-media environments, **localized tampering** (e.g., inpainting-based edits) and **full synthesis** coexist, and these two manipulation modes exhibit distinct artifact characteristics and distributional behaviors. Achieving stable performance across detection, localization, and explanation therefore requires exposure to **multiple synthesis families**, not to maximize quantity, but to capture the heterogeneity of manipulation styles that appear in real social-media data.
>
> **Q2: Whether a small subset of generators could suffice?**
>
> Thank you for bringing up this important point. To examine whether a small subset of generators could suffice, we conducted controlled experiments using UniverFD, selected for its strong cross-generator generalization capability demonstrated in prior work. We retrained UniverFD on reduced subsets of So-Fake-Set, including StyleGAN3 (~30K images), Latent Diffusion (LDM, ~90K images), and their combination (StyleGAN3 + LDM). For each configuration, we sampled real images to match the corresponding fake image count, ensuring balanced training distributions. All other training settings remained fixed. Results are shown in Table 1.
>
> **Table 1: Effect of Reduced Generator Diversity**
> | Training Subset              | So-Fake-Set (Acc %) | So-Fake-OOD (Acc %) |
> |------------------------------|---------------------|----------------------|
> | StyleGAN3                    | 46.8                | 34.2                 |
> | LDM                          | 49.3                | 34.6                 |
> | StyleGAN3 + LDM              | 49.2                | 34.9                 |
> | **Full-set (all families)**     | **84.0**             | **63.8**              |
>
>
> Two observations follow directly from Table 1:
>
> **In-domain accuracy collapses.** When using reduced subsets: the best reduced configuration (StyleGAN3 + LDM: 49.2%) is far below the full multi-family training result (84.0%).
> **OOD accuracy follows the same trend.** The best reduced subset reaches only 34.9%, far below the full-set result of 63.8%.
>
> The results align with the reviewer’s concern. Reduced subsets provide only limited OOD generalization, with clear drops in both in-domain and OOD accuracy. This indicates that limited diversity leads detectors to rely on generator-specific patterns. Full multi-family training instead learns more stable and transferable cues, confirming that synthesis diversity is necessary for reliable performance.
>
> **Q3: Whether diverse generation methods concretely improve generalization?**
>
> The results in Table 1 provide empirical evidence. To complement these findings, we explain why synthesis method diversity in So-Fake leads to stronger generalization.
>
> **First**, So-Fake provides richer supervision beyond binary labels. Pixel-level masks and structured explanations guide models to learn spatially grounded cues shared across generators, reducing reliance on generator-specific artifacts.
>
> **Second**, the dataset reflects realistic social media manipulation patterns. Local tampering and full synthesis follow different creation pipelines but coexist on platforms, requiring detectors to handle mixed, compressed, and heterogeneous content.
>
> **Third**, So-Fake-OOD serves as a rigorous generalization benchmark. Its authentic social media origin and inclusion of unseen commercial generators enable realistic evaluation of cross-domain robustness.
>
> We thank the reviewer for this insightful question, which helped us clarify the rationale behind our design choices.

---

> ### Author Response · Authors · 2025-11-19
> **Response to Reviewer GQQN: W3**
>
> Thank you for bringing up this important point. We address it in two parts: **(Q1) what fundamentally differentiates our social media OOD data from existing academic datasets**, and **(Q2) why these images are more challenging to detect, including whether platform-induced processing traces contribute to this difficulty.**
>
>
> **Q1: What are the differences between these data and other datasets?**
>
> So-Fake-OOD is fundamentally different because it directly uses authentic social media content. Although several recent datasets claim relevance to social media (Table 1 in the main paper), their data collection pipelines differ in crucial ways:
>
> - **SID-Set:** Uses curated academic datasets (COCO, Flickr30k) as real images rather than platform-native content.
> - **TrueFake:** Includes synthetic images sourced from social media but still relies on standard datasets (FFHQ, FORLAB) for real images.
> - **Deepfake-Eval-2024:** Contains only 1,975 images and focuses solely on detection, without a comprehensive multi-task evaluation protocol.
>
> In contrast, So-Fake-OOD collects real images directly from Reddit through the official API, ensuring genuine platform-native characteristics. Unlike standardized collections such as COCO or FFHQ, which provide PNG images at consistent resolutions  (e.g., 512×512), authentic Reddit imagery exhibits substantial heterogeneity: varying resolutions (e.g., 768×1440), multi-stage JPEG re-encoding with platform compression, screenshots, reposted content, and other platform-induced artifacts rarely seen in academic datasets.
>
> These characteristics create a natural domain shift relative to the curated real images in So-Fake-Set (Appendix Figure 8, C1), making So-Fake-OOD the first benchmark to include genuine social media real data for comprehensive evaluation of detection, localization and explanation.
>
>
>
> **Q2: If So-Fake-OOD is harder to detect, what causes the difficulty? Are platform traces responsible?**
>
> Thank you for raising this insightful question. Understanding why social media images behave differently is essential for interpreting the performance gap observed on So-Fake-OOD.
>
> We clarify that So-Fake-OOD is not designed to be artificially difficult, but rather to be **more realistic**, **more truly unseen**, and **more aligned with real deployment conditions.** The observed performance gap reflects a deployment alignment gap and stems from three sources of genuine distributional shift:
>
> 1. **Authentic platform artifacts.** Real images contain genuine social media processing traces such as multi-stage compression, diverse device pipelines, and user edits, which arise organically from platform-native data collection.
>
> 2. **Unseen commercial generators.** Synthetic images come from closed-source commercial models (FLUX, Imagen4, Hidream3) that never appear in training, reflecting realistic deployment where detectors must handle emerging generation tools.
>
> 3. **Deployment-aligned distributions.** So-Fake-OOD combines authentic real images with commercial synthetic content, mirroring the mixture seen on actual social platforms.
>
> To empirically verify that the real-image distribution drives this gap, we conducted a controlled experiment during evaluation. We constructed a hybrid test set by replacing the COCO and CelebA real images (from the So-Fake-Set validation split) with an equal number of Reddit real images, while keeping the fake images unchanged. As shown in Table 1, the model’s accuracy drops sharply (82.4% → 63.3%) on this modified test set, confirming that social-media real images introduce a meaningful domain gap independent of the synthetic content.
>
> **Table 1: Domain gap driven by real-image distribution**
>
> | Method | Real Image Source | Synthetic Image Source | Accuracy (%) |
> |--------|-------------------|------------------------|--------------|
> | UniverFD | COCO + CelebA | So-Fake-OOD | 82.4 |
> | UniverFD | Reddit | So-Fake-OOD | 63.3 |
>
> These results demonstrate that the performance gap reflects the mismatch between academic training distributions and real-world deployment conditions, which So-Fake-OOD is designed to measure. We thank the reviewer for this important question.

---

> ### Author Response · Authors · 2025-11-19
> **Response to Reviewer GQQN: W4, W5, and W6**
>
> **W4: Are there any unique innovations in data generation approach?**
>
>  We appreciate the reviewer’s observation.
>
> Our data generation pipeline focuses on practical realism rather than algorithmic novelty. The goal of Section 3.3 is to reproduce how AI-generated and AI-edited images actually appear on social media platforms, rather than to introduce new synthesis methods.
>
> The contribution therefore lies in the design choices that enable a realistic, multi-task social media benchmark:
>
> **1. Multi-task annotation structure**: So-Fake provides a unified three-way taxonomy (real, tampered, fully synthetic) with pixel-level tampering masks and structured explanations, which together support joint evaluation of detection, localization, and explanation.
>
> **2. Commercial generator integration**: Unlike previous datasets that rely almost exclusively on open-source models, So-Fake-OOD incorporates closed commercial models, reflecting the actual generative tools encountered in real deployments.
>
> **3. Authentic platform processing**: Our tampering pipeline operates directly on Reddit images, preserving natural compression artifacts, repost traces, and heterogeneous resolutions that curated academic datasets typically do not contain.
>
> We hope this clarifies the design rationale and contribution of our generation pipeline.
>
> **W5: The core originality of the method is insufficient.**
>
> Thank you for this comment. We appreciate the opportunity to clarify the intended role and contribution of So-Fake-R1.
>
> **1. Task-oriented baseline rather than methodological novelty**
> So-Fake-R1 is intentionally built on established components such as cold-start initialization and GRPO. The aim is to provide a reliable unified baseline for the three-task setting of So-Fake (detection, localization, and explanation), not to introduce a new algorithm. A stable and representative baseline is essential for benchmarking.
>
> **2. Reinforcement learning-based unification of three heterogeneous tasks**
> So-Fake-R1 applies reinforcement learning to jointly optimize detection, localization, and explanation—tasks that are typically addressed separately in prior work. These outputs involve heterogeneous formats (class labels, bounding boxes, structured text) that are difficult to combine with standard supervised losses. The reward-based formulation offers a unified optimization framework that enables cohesive multi-task evaluation on So-Fake.
>
> **3. Empirical validation of effectiveness**
> Tables 2 and 3 in the main paper show that So-Fake-R1 performs well across all three tasks while producing interpretable outputs. This demonstrates that the reinforcement learning formulation is effective and well-suited to the benchmark.
>
> We hope this clarifies the intended purpose of So-Fake-R1 within our work.
>
> **W6: Cross-dataset experiments**
>
> Thank you for this suggestion. We agree that cross-dataset evaluation is essential for demonstrating generalization. In our work, we have incorporated such evaluation through multiple test sets with different data distributions.
>
> Specifically, beyond So-Fake-OOD, we tested So-Fake-R1 on SID-Set, which covers different AI generators while maintaining a similar task structure. The model achieves strong performance on SID-Set (93.4% overall accuracy, Table 5), demonstrating effective generalization to unseen generators and validating the robustness of our approach across AIGC detection benchmarks.
>
> Regarding the suggested experiment on Photoshop-based datasets like CAT-Net, we identify two key compatibility challenges. First, these datasets focus on traditional splicing/copy-move manipulations with binary labels and masks, lacking the AI-generated content, AI-editing samples, and textual rationales that So-Fake-R1 is designed to handle. Second, the task formulation differs fundamentally: CAT-Net addresses classical image forensics, while our work targets modern AI-generated and AI-edited content prevalent on social media platforms. Training on CAT-Net would require removing core components of our three-task framework, essentially creating a different method.
>
> Given these differences, So-Fake-OOD was specifically designed to evaluate cross-domain generalization within our target setting. It includes authentic social media images and multiple commercial generators unseen during training, providing a rigorous test of OOD performance for AI-generation forensics. This design choice allows us to isolate distribution shift from task incompatibility.
>
> We acknowledge that exploring traditional manipulation datasets remains valuable future work and would provide additional insights into cross-paradigm generalization. We plan to extend So-Fake by incorporating Photoshop-based manipulation types in future iterations, creating a more comprehensive benchmark that spans both classical forensics and modern AIGC detection.

---

> ### Author Response · Authors · 2025-11-22
> **W6: Cross-dataset experiments**
>
> We agree with the reviewer that verifying performance on traditional manipulation benchmarks (e.g., CAT-Net protocols) is essential for demonstrating broad generalization.
>
> To address this, we conducted an additional experiment examining cross-paradigm transfer. Since the datasets utilized by CAT-Net lack the textual rationales required by our architecture, we utilized the **MMTD-Set** [1] as a bridge. This dataset is built upon the same benchmarks evaluating CAT-Net but provides the necessary triplet supervision (image, mask, text), allowing us to evaluate So-Fake-R1 on classical splicing/copy-move distributions without altering our core framework.
>
> **Experimental Settings.** To validate transferability under resource constraints, we devised a specific adaptation protocol for this rebuttal:
> -   **Data Construction:** We randomly sampled a balanced subset of **5,000 examples** (2,500 real and 2,500 tampered) from MMTD-Set for Supervised Fine-Tuning (SFT).
> -   **Task Adaptation:** To align with the binary nature of traditional forensics, we modified the classification head of So-Fake-R1 from three-way (Real/Tampered/Full Synthetic) to **binary (Real/Tampered)**.
> -   **Reward Re-calibration:** Crucially, for the subsequent RL stage, we **redefined the classification reward** to optimize for this binary distinction, while keeping the localization and explanation rewards unchanged. We then performed a short RL phase of **1,000 steps**, while keeping the remaining hyperparameters identical to those in the main paper.
>
> **Table 1: Detection Accuracy**
> | Method        | CASIA1+ | IMD2020 | Columbia |
> |---------------|---------|---------|----------|
> | SPAN          | 60.0    | 70.0    | 87.0     |
> | ManTraNet     | 52.0    | 75.0    | 95.0     |
> | CAT-Net       | 88.0    | 68.0    | 90.0     |
> | PSCC-Net      | 90.0    | 67.0    | 78.0     |
> | FakeShield    | 95.0    | 83.0    | 98.0     |
> | Ours (So-Fake-R1)| 94.7 | 74.5 | 98.2 |
>
> **Table 2: Localization IOU**
> | Method        | CASIA1+ | IMD2020 | Columbia |
> |---------------|---------|---------|----------|
> | SPAN          | 11.0    | 9.0     | 14.0     |
> | ManTraNet     | 9.0     | 10.0    | 4.0      |
> | OSN           | 47.0    | 38.0    | 58.0     |
> | CAT-Net       | 44.0    | 14.0    | 8.0      |
> | PSCC-Net      | 36.0    | 22.0    | 64.0     |
> | FakeShield    | 54.0    | 50.0    | 67.0     |
> | Ours (So-Fake-R1) | 49.3 | 46.7 | 63.4 |
>
> Despite the abbreviated training schedule and limited data, the results in Table 1 and 2 demonstrate that So-Fake-R1 achieves performance comparable to established forensic methods, effectively validating its transferability beyond the AIGC domain. We acknowledge that the current performance is constrained by the strict rebuttal timeline, and we expect further improvements with a full training. Moving forward, rather than relying solely on existing academic benchmarks, we plan to generate a dataset of classical manipulations (e.g., copy-move, splicing) applied specifically to social media imagery, which will better align traditional forensics with our target real-world setting.
>
> We sincerely thank the reviewer for raising this point. This experiment highlights a critical aspect of generalization that was not covered in the initial submission, and addressing it has significantly broadened the scope and robustness of our study.
>
> [1] Xu, Zhipei et al. “FakeShield: Explainable Image Forgery Detection and Localization via Multi-modal Large Language Models.” ICLR, 2025.

---

> > ### Comment · Reviewer_GQQN · 2025-11-26
> >
> > I appreciate the authors' rebuttal, where the additional experiments and explanations provided have addressed some of my concerns. However, I still maintain that the core contributions of this paper, i.e., specifically its dataset and methodology, exhibit certain limitations:
> > - Dataset: The claim that So-Fake possesses distinct characteristics not found in existing datasets lacks sufficient rigor (e.g., what constitutes "real social-media posting and manipulation patterns"? What specific differences does "Expanded Content Coverage" offer compared to existing datasets?).
> > - Methodology: The reinforcement learning framework and GRPO algorithm employed in So-Fake-R1 are not original. Treating elements such as "Commercial generator integration" and "Authentic platform processing" as core innovations is relatively weak.
> >
> > Therefore, I stand by my initial evaluation.

---

> ### Author Response · Authors · 2025-11-26
> **Response to Reviewer GQQN**
>
> We are pleased that our previous responses have addressed several of your initial concerns. Regarding the two remaining concerns, we note that relevant evidence has been provided in our earlier replies. We appreciate this opportunity to consolidate the key findings here for clarity.
>
> **Concern 1: Dataset**
>
> **(1) Distributional differences (Response W3, Q1):**
>
> So-Fake-OOD uses platform-native Reddit images with variable resolutions and multi-stage compression, unlike curated academic datasets (COCO, FFHQ) with fixed resolution and single-stage compression.
>
> | Dataset | Real Image Source | Key Difference |
> |-|-|-|
> | Existing | COCO, FFHQ, etc. | Curated, fixed resolution |
> | **So-Fake-OOD** | **Reddit (API)** | **Platform-native, multi-task** |
>
> **(2) Empirical validation of platform characteristics (Response W3, Q2):**
>
> To isolate the impact of platform-specific artifacts, we conducted a controlled experiment with UniverFD. We replaced academic real images (COCO/CelebA) with Reddit images in the test set, while keeping the synthetic images identical:
>
> | Real Image Source | Synthetic Image Source | Accuracy |
> |-|-|-|
> | COCO + CelebA  | So-Fake-OOD synthetic | 82.4% |
> | Reddit| So-Fake-OOD synthetic | 63.3% |
>
> Finding: **19.1%** accuracy drop. Since synthetic data was fixed, this collapse is exclusively driven by the domain gap in real social media images .
>
> **(3) Empirical validation of semantic diversity (Response W1, Q3):**
>
> To validate whether expanded content coverage genuinely improves cross-category robustness, we conducted a controlled experiment using CNNSpot with identical training data size, varying only the semantic scope.
>
> | Training Setting | Face AUROC | Non-Face AUROC |
> |------------------|------------|----------------|
> | Face-only training | 0.960 | 0.562 |
> | Multi-category (12 classes) | 0.897 | 0.848 |
>
> **Finding:** Face-only training collapses on non-face content (0.562 AUROC) while multi-category training achieves significant improvement (→0.848), validating that semantic breadth prevents overfitting.
>
> **Concern 2: Methodology**
>
> We clarify that "Commercial generator integration" and "Authentic platform processing" were discussed in Response W4 as **dataset design choices** for benchmark realism, not as algorithmic innovations. We address both the dataset design rationale and method positioning below:
>
> **(1) Benchmark design rationale (Response W4):**
>
> Our pipeline focuses on practical realism: (1) multi-task annotations with pixel-level masks, (2) commercial generators reflecting deployment scenarios, (3) authentic platform processing preserving compression artifacts. These are benchmark design choices, not methodological innovations.
>
> **(2) Method contribution and empirical validation:**
>
> **Method contribution (Response W5):** So-Fake-R1 is a **unified baseline** for the three-task benchmark. The reinforcement learning formulation provides an optimization framework for various outputs (class labels, bounding boxes, structured text) that are difficult to combine with standard supervised losses.
>
> **Cross-domain validation (Response W6, Additional):**
>
> To validate transferability beyond the AIGC domain, we trained So-Fake-R1 on MMTD-Set (classical splicing/copy-move forensics) with task adaptation from three-way to binary classification. Despite abbreviated training (5K samples, 1K RL steps), results are comparable to established forensic method:
>
> **Detection Accuracy:**
> | Method | CASIA1+ | IMD2020 | Columbia |
> |-|-|-|-|
> | CAT-Net | 88.0% | 68.0% | 90.0% |
> | **So-Fake-R1** | **94.7%** | **74.5%** | **98.2%** |
>
> **Localization IOU:**
> | Method | CASIA1+ | IMD2020 | Columbia |
> |-|-|--|-|
> | CAT-Net | 44.0 | 14.0 | 8.0 |
> | **So-Fake-R1** | **49.3** | **46.7** | **63.4** |
>
> **Finding:** So-Fake-R1 achieves **94.7% detection accuracy** and **49.3 IOU** on CASIA1+, demonstrating effective transferability to traditional forensics.
>
> **Summary:**
>
> The evidence above consolidates three controlled experiments directly addressing the two concerns:
>
> | Experiment | Key Finding | Response |
> |-|-|-|
> | Platform-native data analysis | **19.1% domain gap** quantification | W3, Q2 |
> | Semantic diversity validation | **28.6%  improvement**  in cross-category generalization | W1, Q3 |
> | Cross-domain validation | **94.7% detection, 49.3 IOU** on classical forensics | W6, Additional |
>
> These experiments demonstrate empirical rigor through controlled comparisons, systematic ablations, and cross-domain validation. We thank you for the opportunity to consolidate this evidence. Your feedback has helped us articulate our contributions more clearly and strengthen the empirical foundation of our work. Should you have additional concerns, we would be happy to engage in further discussion.

---

### Official Review · Reviewer_QhmF · 2025-10-28

**Soundness:** 3
**Presentation:** 4
**Contribution:** 4
**Rating:** 8
**Confidence:** 5

**Summary:**

This paper studies the task of forgery analysis on static images, including detection, localization, and anomaly explanation. Targeting the social media domain, where AI-generated images are widespread, the authors propose **So-Fake-Set** and **So-Fake-OOD**, along with a reinforcement learning-based baseline method named **So-Fake-R1**. Specifically, to address the scarcity of social media image data, the authors collect and construct a large-scale dataset and a challenging OOD benchmark tailored for social media images, which holds significant potential for advancing the community. In addition, the introduction of reinforcement learning into the forgery analysis task achieves impressive results.

**Strengths:**

1. The proliferation of AI-generated images on social media is indeed one of the pressing issues in today’s society. The proposed dataset effectively alleviates the data scarcity problem in this area and carries significant research value at the frontier of the field.
2. With a simple reward function design, **So-Fake-R1** achieves state-of-the-art performance, opening up new possibilities for applying reinforcement learning to forgery image analysis tasks.
3. The paper is clearly written, and the figures and visual analyses are comprehensive and easy to follow.

**Weaknesses:**

As a benchmark, the evaluation should be expanded to include as many Vision-Language Models (VLMs) as possible — for example, **DeepSeek-VL2** and several closed-source Multimodal Large Language Models (MLLMs) such as **GPT**. In addition, experiments should also be conducted across different model sizes to provide a more comprehensive assessment.

**Questions:**

Please refer to the *Weakness* section for detailed explanations.

---

> ### Author Response · Authors · 2025-11-19
> **Response to Reviewer QhmF**
>
> We appreciate your valuable suggestion.
>
> To address this concern, we expanded our evaluation to include multiple Vision–Language Models across different parameter scales, including models such as DeepSeek-VL2-tiny and Qwen2.5-Omni-3B, alongside the closed-source GPT-4o. These models represent the largest open-source VLMs we could reliably finetune within the available rebuttal time and compute budget. For comparability, all models are evaluated on a balanced test subset of 900 randomly sampled images per split (300 per class: real, tampered, fully synthetic) from both So-Fake-Set and So-Fake-OOD.
>
> **Table 1: Results on So-Fake-Set**
> | Method                       | Detection Acc | Detection F1 | Localization IOU | Localization F1 |
> |------------------------------|---------------|--------------|-------------------|------------------|
> | DeepSeek-VL2-tiny (finetune) | 75.5          | 73.8         | 37.5              | 43.0             |
> | Qwen2.5-Omni-3B (finetune)   | 88.4          | 87.4         | 43.1              | 55.2             |
> | GPT-4o (few-shot)            | 51.2          | 50.3         | 1.9               | 3.2              |
> | So-Fake-R1         | 93.3        | 93.0       | 48.5            | 63.7           |
>
>
> **Table 2: Results on So-Fake-OOD**
> | Method                       | Detection Acc | Detection F1 | Localization IOU | Localization F1 |
> |------------------------------|---------------|--------------|-------------------|------------------|
> | DeepSeek-VL2-tiny (finetune) | 63.3          | 63.0         | 27.5              | 33.9             |
> | Qwen2.5-Omni-3B (finetune)   | 70.7          | 69.8         | 41.5              | 47.0             |
> | GPT-4o (few-shot)            | 45.1          | 44.8         | 1.1               | 2.5              |
> | So-Fake-R1         | 76.5       | 75.2      | 47.7           | 59.0           |
>
> As shown in Tables 1 and 2, after finetuning on the So-Fake-Set training split, both DeepSeek-VL2-tiny and Qwen2.5-Omni-3B achieve reasonably strong detection and localization performance on the So-Fake-Set validation set and on So-Fake-OOD, indicating that VLM backbones can benefit from our setting. In contrast, GPT-4o is evaluated only as a supplementary reference using a unified **few-shot prompting protocol**. We first try a simple direct prompt, which leads GPT-4o to classify almost all images as real. We then design a more careful few-shot prompt, which produces more varied outputs but still yields clearly inferior detection accuracy and almost no usable localization. Notably, GPT-4o shows a systematic bias toward the “real” label, consistent with widely reported tendencies of multimodal LLMs to avoid over-claiming synthetic or manipulated content.
>
> Together, these findings indicate that while finetuned VLMs can adapt to the fine-grained signals required for forgery analysis, an off-the-shelf model like GPT-4o remains unsuitable for this task under the current prompting setup. We view this as an encouraging direction and plan to explore more effective prompting and adaptation strategies in future work.

---

> > ### Comment · Reviewer_QhmF · 2025-11-27
> >
> > Thank you for your efforts and detailed responses. I would like to maintain the original score.

---

> > > ### Author Response · Authors · 2025-11-27
> > > **Sincere Thanks**
> > >
> > > Dear Reviewer QhmF,
> > >
> > > We sincerely appreciate your constructive feedback and support. The additional experiments have significantly strengthened the empirical comprehensiveness of our work, and we will ensure these results are incorporated into the final version.
> > >
> > > Best regards,
> > >
> > > Authors

---

### Official Review · Reviewer_4LoV · 2025-11-01

**Soundness:** 3
**Presentation:** 3
**Contribution:** 3
**Rating:** 6
**Confidence:** 3

**Summary:**

This paper introduces So-Fake, a large-scale benchmark for social media image forgery detection, and a unified baseline model So-Fake-R1. So-Fake provides diverse, realistic, and out-of-distribution data that reflect real-world social media conditions. Built upon this benchmark, So-Fake-R1 employs a two-stage training framework combining supervised fine-tuning and reinforcement learning (GRPO) to jointly perform these three tasks with structured reasoning outputs.

**Strengths:**

1.	This paper is easy to follow.
2.	Large-scale dataset. The proposed benchmark is significantly larger and more diverse than existing datasets, covering 12 semantic categories from real-world social media sources
3.	Well-designed OOD setting. So-Fake-OOD explicitly uses commercial generators that do not overlap with those in the training set, effectively simulating real-world scenarios where models must handle unseen generative tools.

**Weaknesses:**

Since So-Fake-OOD real samples come from Reddit and the training set aggregates from COCO and other sources, there might be near-duplicate or visually similar images across domains. The paper should report duplicate detection to ensure strict separation between training and OOD evaluation sets.

**Questions:**

See weaknesses

---

> ### Author Response · Authors · 2025-11-19
> **Response to Reviewer 4LoV**
>
> Thank you for highlighting this point. We appreciate the reviewer’s perspective, and we address it from two complementary angles as outlined below.
> 1. **Disjoint collection mechanisms**: We emphasize that the real-image components of **So-Fake-Set** and **So-Fake-OOD** are **fully disjoint by construction**. So-Fake-Set uses only static, version-controlled academic datasets (COCO, Flickr30k, CelebA, FFHQ, OpenForensics, and OpenImages-v7), all released between 2014–2022 through curated annotation pipelines. In contrast, **So-Fake-OOD** real images are collected directly from **Reddit** using temporal and ranking-based filters, sampling from 11 subreddits with posts exclusively from **2024–2025**. Because the two pipelines differ in **platform, time period, data acquisition procedure, and content dynamics**, their distributions do not overlap, so accidental near-duplicates are unlikely.
> 2. **Empirical Duplicate Detection：** To verify this claim, we conducted comprehensive similarity analysis using DINOv3 (ViT-L/16), a state-of-the-art self-supervised vision model widely adopted for duplicate detection tasks.
>
>     **Experimental Setup:** To ensure a rigorous and scalable analysis, we evaluate duplicate similarity under the following configuration:
>     -   **Data sampling:** Due to computational constraints during rebuttal, we randomly sample 10% of the real images from each split (seed = 42), resulting in **65,000** sampled real images from So-Fake-Set (650K total) and **3,300** from So-Fake-OOD (33K total).
>     -   **Similarity metric:** We compute cosine similarity from DINOv3 embeddings and use a threshold of 0.9.
>
>     **Results:** The experiment identified **19 high-similarity pairs** across the two subsets. We manually inspected all 19 pairs, with representative examples now included in **Appendix C.5 of the revised manuscript**. Importantly, these cases reflect **topic-level similarity rather than actual duplicates**. For example:
>     -  **Pair #1:** Two different photographs of Moraine Lake, taken from comparable viewpoints but exhibiting distinct lighting, composition, and seasonal conditions.
>     -   **Pair #2:** The same vehicle model photographed in different settings.
>
>     Although these pairs achieve high DINOv3 similarity scores, visual examination confirms that they are **clearly different images depicting related subjects or scenes**, which is expected when comparing large-scale, diverse image collections.
> We will incorporate the complete duplicate detection analysis and corresponding visual examples in the final version. We thank the reviewer for the helpful feedback, which strengthened this part of our work.

---

> ### Comment · Area_Chair_GMV5 · 2025-11-28
> **Rebuttal Review Request**
>
> Dear Reviewers,
>
> Thank you for your time and thoughtful feedback on this manuscript.
>
> The authors have now submitted their rebuttal. If you haven’t already, we kindly ask you to review their responses and consider whether your concerns have been adequately addressed.
>
> Best regards,
>
> AC

---

### Author Response · Authors · 2025-11-29

Dear PCs, SACs, and ACs,

 Thank you for taking on this submission under these challenging circumstances. We have provided detailed responses to all reviewer concerns during the rebuttal period.

We appreciate your careful consideration.

Best regards,

Authors

---

### Meta-Review · Area_Chair_3s8e · 2026-01-08

**Summary:**

This work proposes a benchmark for social media image forgery detection. The authors propose So-Fake dataset consisting of 2 million training images and 100K out-of-domain test images, as well as So-Fake-R1 framework that uses reinforcement learning to encourage interpretable visual rationales. Reviewers recognized the contribution of the dataset and benchmark to the community, the rigor of the proposed benchmark and evaluation, and the extensive experiment analysis. Concerns are raised on the potential duplicate between training and testing, benchmarking on more base models, unique innovations on dataset curation and method design, lack of cross-dataet experiments, insufficient ablation on scalability, and potential domain limitations.

**Reviewer Concerns:**

Reviewer 4LoV's concern on potential duplicate between training and testing is addressed during rebuttal by explaining the disjoint collection mechanisms and conducting an empirical duplicate detection study.

Reviewer QhmF's concern on benchmarking more models has been addressed during the rebuttal.

For Reviewer GQQN, some concerns are addressed during the rebuttal, while the concern on distinct characteristics of the proposed dataset and novelty of the proposed RL approach is not completely solved.

Reviewer npdH's concern on scalability experiments is addressed by an additional experiment, while the concern on domaion limitation is not fully addressed (the authors argued that the current dataset is diverse and acknowledged that the single data source is a limitation of the current work).

**Reviewer Scores:**

Reviewer 4LoV gave the initial score of borderline accept. Given that the reviewer's only concern on potential duplicate has been addressed during rebuttal, it is likely that reviewer 4LoV will increase the score to weak accept, or at least keep the original borderline accept score.

Reviewer QhmF gave the initial score of accept, and indicated during discussion period that the reviewers addressed his/her concerns and he/she will keep the original score.

Reviewer GQQN gave the initial score of borderline reject, and indicated during discussion period that he/she will maintain the original score.

Reviewer npdH gave the initial score of borderline accept, and is likely to maintain or lower the score (to borderline reject) after rebuttal.

The area chair shares the concerns of Reviewer GQQN and npdH on insufficient contribution and novelty, and limited diversity due to single data source. These two concerns are not fully addressed during rebuttal

---

### Decision · Program_Chairs · 2026-01-26

Reject